# The hidden role of dissolved organic carbon in the biogeochemical cycle of carbon in modern redox-stratified lakes

Robin Havas[a,*], Christophe Thomazo[a,b], Miguel Iniesto[c], Didier Jézéquel[d], David Moreira[c], Rosaluz Tavera[e], Jeanne Caumartin[f], Elodie Muller[f], Purificación López-García[c], Karim Benzerara[f]

[a] Biogéosciences, CNRS, Université de Bourgogne Franche-Comté, 21 000 Dijon, France

[b] Institut Universitaire de France, 75005 Paris, France

[c] Ecologie Systématique Evolution, CNRS, Université Paris-Saclay, AgroParisTech, 91190 Gif-sur-Yvette, France

[d] IPGP, CNRS, Université Paris Cité, 75005 Paris, and UMR CARRTEL, INRAE & USMB, France

[e] Departamento de Ecología y Recursos Naturales, Universidad Nacional Autónoma de México, México

[f] Sorbonne Université, Muséum National d'Histoire Naturelle, CNRS, Institut de Minéralogie, de Physique des Matériaux et de Cosmochimie (IMPMC), 75005 Paris, France.

[*] *Correspondence to*: Robin Havas (robin.havas@gmail.com)

*Keywords: Carbon cycle; isotopic fractionation; DOC; Precambrian analogues*

**Abstract.** The dissolved organic carbon (DOC) reservoir plays a critical role in the C cycle of marine and freshwater environments because of its size and implication in many biogeochemical reactions. Although it is poorly constrained, its importance in ancient Earth's C cycles is also commonly invoked. Yet DOC is rarely quantified and characterized in modern stratified analogues. In this study, we investigated the DOC reservoirs of four redox-stratified alkaline crater lakes in Mexico. We analyzed the concentrations and isotopic compositions of DOC throughout the four water columns and compared them with existing data on dissolved inorganic and particulate organic C reservoirs (DIC and POC). The four lakes have high DOC concentrations with great variability between and within the lakes (averaging $2 \pm 4$ mM; 1SD, n=28; i.e. from $\sim 15$ to 160 times the amount of POC). The $\delta^{13}C_{DOC}$ signatures also span a broad range of values from -29.3 to -8.7 ‰ (with as much as 12.5 ‰ variation within a single lake). The prominent DOC peaks (up to 21 mM), together with their associated isotopic variability, are interpreted as reflecting oxygenic and/or anoxygenic primary productivity through the release of excess fixed carbon in three of the lakes (La Alberca de los Espinos, La Preciosa, and Atexcac). By contrast, the variability of [DOC] and $\delta^{13}C_{DOC}$ in the case of Lake Alchichica is mainly explained by the partial degradation of organic matter and accumulation of DOC in anoxic waters. The DOC records detailed metabolic functions such as active DIC-uptake and DIC-concentrating mechanisms, which cannot be inferred from DIC and POC analyses alone, but which are critical to the understanding of carbon fluxes from the environment to the biomass. Extrapolating our results to the geological record, we suggest that anaerobic oxidation of DOC may have caused the very negative C isotope excursions in the Neoproterozoic. It is however unlikely that a large oceanic DOC reservoir could overweigh the entire oceanic DIC reservoir. This study demonstrates how the analysis of DOC in modern systems deepens our understanding of the C cycle in stratified environments and helps to set boundary conditions for the Earth's past oceans.

## 1. INTRODUCTION

Dissolved organic carbon (DOC) is a major constituent of today's marine and freshwater environments (e.g. Ridgwell and Arndt, 2015; Brailsford, 2019). It is an operationally defined fraction of aqueous organic carbon within a continuum of organic molecules spanning a broad range of sizes, compositions, degrees of reactivity, and bioavailability (Kaplan et al., 2008; Hansell, 2013; Beaupré, 2015; Carlson and Hansell, 2015; Brailsford, 2019). Oceanic DOC is equivalent to the total amount of atmospheric carbon (Jiao et al., 2010; Thornton, 2014) and represents the majority of freshwater organic matter (Kaplan et al., 2008; Brailsford, 2019). The DOC reservoir is (i) at the base of many trophic chains (Bade et al., 2007; Hessen and Anderson, 2008; Jiao et al., 2010; Thornton, 2014), (ii) key in physiological and ecological equilibria (Hessen and Anderson, 2008) and (iii), has a critical role for climate change as a long-term C storage reservoir (Jiao et al., 2010; Hansell, 2013; Thornton, 2014; Ridgwell and Arndt, 2015). Although isotopic signatures are a powerful and widespread tool in biogeochemical studies, the use of DOC isotopes has been relatively limited owing to technical difficulties (Cawley et al., 2012; Barber et al., 2017). Radioisotopes or labeled stable isotopes of DOC have been used to date and retrace DOC compounds in diverse aquatic environments (e.g. Repeta and Aluwihare, 2006; Bade et al., 2007; Kaplan et al., 2008; Brailsford, 2019). Studies featuring natural abundances of DOC stable isotope data (i.e. $\delta^{13}C_{DOC}$) mainly used them to discriminate between different source endmembers (e.g. terrestrial vs. autochthonous) (e.g. Cawley et al., 2012; Santinelli et al., 2015; Barber et al., 2017). After a pioneer study by Williams and Gordon (1970), few studies have used natural DOC stable isotope compositions to explore processes intrinsically related to its production and recycling. Recently, Wagner et al. (2020) reaffirmed the utility of stable isotopes to investigate DOC biosynthesis, degradation pathways, and transfer within the foodweb.

Several studies have suggested a significant role for the DOC reservoir throughout geological time, when it would have been much larger in size and impacting various phenomena, including: the regulation of climate and glaciations during the Neoproterozoic (e.g. Peltier et al., 2007), the paleoecology of Ediacaran Biota and its early complex life forms (e.g. Sperling et al., 2011), the oxygenation of the ocean through innovations of eukaryotic life near the Neoproterozoic-Cambrian transition (e.g. Lenton and Daines, 2018), and the perturbation of the C cycle recorded in $\delta^{13}C$ sedimentary archives from the Neoproterozoic to the Phanerozoic (e.g. Rothman et al., 2003; Fike et al., 2006; Sexton et al., 2011; Ridgwell and Arndt, 2015).

The contribution of DOC reservoirs to the past and modern Earth's global climate and biogeochemical cycles remains poorly constrained (Jiao et al., 2010; Sperling et al., 2011; Dittmar, 2015; Fakhraee et al., 2021) and the existence and consequences of a large ancient oceanic DOC are still debated (e.g. Jiang et al., 2010, 2012; Ridgwell and Arndt, 2015; Li et al., 2017; Fakhraee et al., 2021). Thus, in addition to modeling approaches (e.g. Shi et al., 2017; Fakhraee et al., 2021), the understanding of DOC-related processes in the past anoxic and redox-stratified oceans (Lyons et al., 2014; Havig et al., 2015; Satkoski et al., 2015) should rely on the characterization of DOC dynamics in comparable modern analogues (Sperling et al., 2011). Although many studies have explored the C cycle of modern redox-stratified environments (e.g. Crowe et al., 2011; Kuntz et al., 2015; Posth et al., 2017; Schiff et al., 2017; Havig et al., 2018; Cadeau et al., 2020; Saini et al., 2021; Petrash et al., 2022), very few have analyzed DOC and even fewer have measured its stable isotope signature (Havig et al., 2018).

In this study, we characterize the DOC reservoir of four modern redox-stratified alkaline crater lakes from the Trans-Mexican Volcanic Belt (Ferrari et al., 2012) and its role within the C cycle of these environments. We report

DOC concentration and isotopic composition at multiple depths in the four water columns, and discuss these results in the context of physico-chemical parameters (temperature, dissolved oxygen, chlorophyll a, and nutrient concentrations), and the isotopic composition of dissolved inorganic and particulate organic carbon (DIC, POC), all measured in the same lakes and from the same water samples as in Havas et al. (2023). The four lakes show distinct water chemistries, along an alkalinity/salinity gradient (Zeyen et al., 2021), with diverse planktonic microbial communities (Iniesto et al., 2022; Havas et al., 2023). These characteristics allow us to examine the effect of specific environmental and ecological constraints on the production and recycling of DOC in redox stratified environments. We then present how the analysis of DOC deepens our understanding of the C cycle in these lakes, compared to more classical DIC and POC analyses. Finally, the production and fate of the DOC reservoir in these modern analogues is used to discuss the potential role of DOC in past perturbations of the sedimentary C isotope record from the Neoproterozoic and Phanerozoic.

## 2. SITE DESCRIPTION

The main characteristics of the geological, climatic and limnological context of the lakes under study are presented here, but a more detailed description is available in Havas et al. (2023).

The four lakes are volcanic maars formed after phreatic, magmatic and phreatomagmatic explosions, and are located in the Trans-Mexican Volcanic Belt (TMVB, Fig. 1). The first lake, La Alberca de los Espinos, is located at the margin of the Zacapu tectonic lacustrine basin in the Michoacán-Guanajuato Volcanic Field (MGVF), in the western-central part of the TMVB (Fig. 1). The other three (La Preciosa, Atexcac, and Alchichica) are located within the same zone (~ 50 km²) of the Serdan-Oriental Basin (SOB), in the easternmost part of the TMVB (Fig. 1). La Alberca, with a temperate semi-humid climate, is predominantly underlain by andesitic rocks (Siebe et al., 2012, 2014). By contrast, Alchichica shows much higher evaporation than precipitation rates, reflecting the temperate sub-humid to temperate arid climate experienced by the SOB lakes (Silva-Aguilera et al., 2022). These lakes overlie calcareous and basaltic/andesitic basement rocks (Carrasco-Núñez et al., 2007; Chako Tchamabé et al., 2020).

These variations in geological context and hydrological processes generate a gradient of water chemical compositions, where salinity, alkalinity and DIC increase in the following order: (i) Lake La Alberca, (ii) La Preciosa, (iii) Atexcac, and (iv) Alchichica (Zeyen et al., 2021). The four lakes are alkaline with pH values around 9. Under these conditions, DIC is composed of $HCO_3^-/CO_3^{2-}$ ions with minor amounts of $CO_{2(aq)}$ (< 0.5 %). This favors the precipitation of microbialite deposits, which are found in the four systems but more abundantly as alkalinity increases (Zeyen et al., 2021).

The four lakes are defined as warm monomictic with anoxic conditions prevailing in the bottom waters during most of the year (i.e. one mixing period per year, during winter; Armienta et al., 2008; Macek et al., 2020; Havas et al., 2023). They are all "closed lakes" with no inflow or outflow of surficial waters and are thus fed by rain and groundwater only.

Atexcac is the most oligotrophic of the three SOB lakes (Lugo et al., 1993; Vilaclara et al., 1993; Sigala et al., 2017). Chlorophyll a data from May 2019 (Fig. 2), based on mean and maximum value categories (OECD, 1982), indicate ultra-oligotrophic conditions for Atexcac (≤ 1 and 2 µg/L, respectively), oligotrophic for Alchichica (≤ 2

and 6 µg/L, respectively), intermediate between oligo- and mesotrophic for La Alberca (≤ 3 and 4.5 µg/L, respectively) and "low" mesotrophic for La Preciosa (≤ 3 and 9 µg/L, respectively). Total dissolved P concentrations from May 2019 show similar values for the three SOB lakes close to the surface (increasing in the anoxic zone of Alchichica) but much higher values for La Alberca (Havas et al., 2023). This pattern was observed during previous sampling campaigns (Zeyen et al., 2021). La Alberca is surrounded by more vegetation, which could favor the input of nutrients to this lake. La Preciosa and La Alberca are thus the least oligotrophic of the four lakes. Importantly, although differences in trophic status exist between the four lakes, they are more oligotrophic than eutrophic.

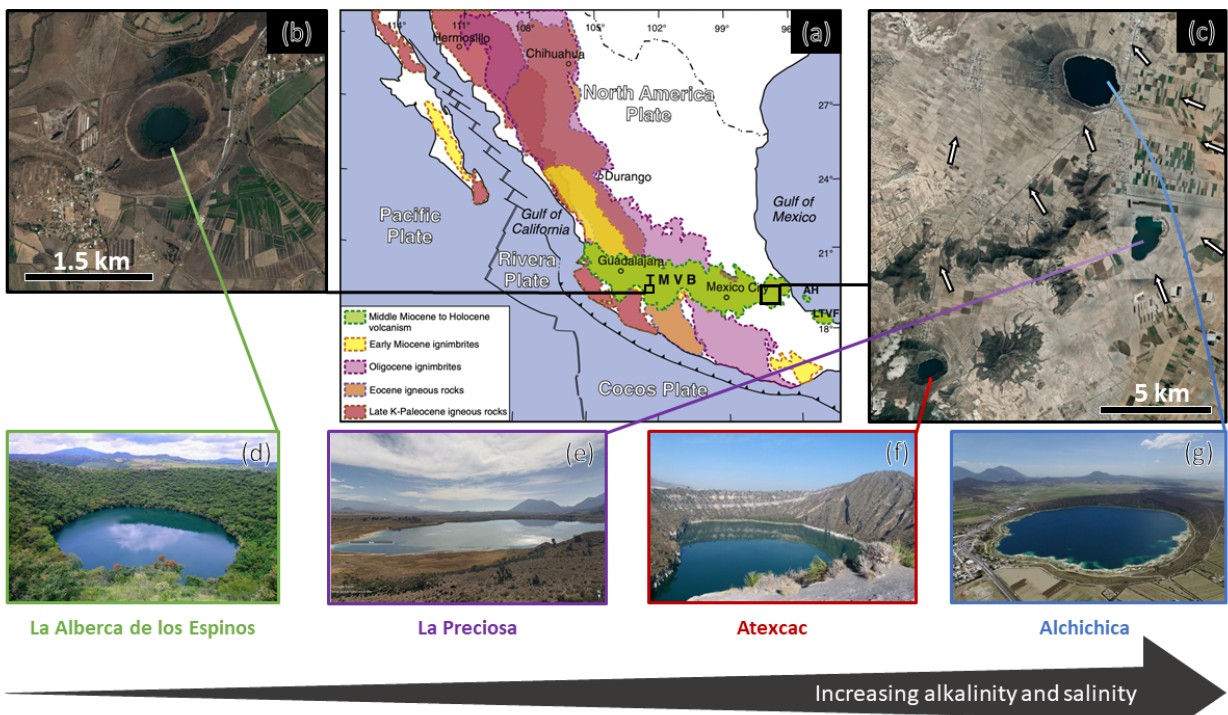

Figure 1. Geographical location and photographs of the four crater lakes. (a) Geological map from Ferrari et al. (2012) with black squares showing the location of the four studied lakes within the Trans-Mexican Volcanic Belt (TMVB). (b, c) Close up © Google Earth views of La Alberca de los Espinos and the Serdan-Oriental Basin (SOB). The white arrows represent the approximate groundwater flow path (based on Silva-Aguilera, 2019). (d-g) Photographs of the four lakes (d from © Google Image ['enamoratedemexicowebsite'], e from © Google Earth street view, and g from © 'Agencia Es Imagen'). Figure from Havas et al. (2023).

## 3. METHOD

### 3.1. Sample Collection

All samples were collected in May 2019. Samples for DOC analyses were collected at different depths from the surface to the bottom of the water columns, particularly where the physico-chemical parameters showed pronounced variation (e.g. at the chemocline and turbidity peaks; Fig. 2 and Table 1). Water samples were collected with a Niskin bottle. For comparison with DIC and POC data, the DOC was analyzed on the same Niskin sampling as in Havas et al. (2023), except where indicated (Fig. 4; Tables 1 and 2). Analyses of DOC and major, minor, and

trace ions were carried out after water filtration at 0.22 μm, directly in the field with Filtropur S filters pre-rinsed with lake water. Details about the sampling procedure and analysis of the physico-chemical parameters, as well as DIC and POC measurements, are reported in Havas et al. (2023).

### 3.2. Dissolved organic carbon (DOC) concentration and isotope measurements

Filtered solutions were acidified to a pH of ~1-2 to degas all the DIC and leave DOC as the only C species in solution. The bulk DOC was analyzed directly from the acidified waters (i.e. all organic C molecules smaller than 0.22 μm). Bulk concentration was measured with a Vario TOC at the Laboratoire Biogéosciences (Dijon), calibrated with a range of potassium hydrogen phthalate (Acros®) solutions. Before isotopic analysis, DOC concentration of the samples was adjusted to match international standards at 5 ppm (USGS 40 glutamic acid and USGS 62 caffeine). Isotopic compositions were measured at the Laboratoire Biogéosciences using an IsoTOC (Elementar, Hanau, Germany), running under He-continuous flow and coupled with an IsoPrime stable isotope ratio mass spectrometer (IRMS; Isoprime, Manchester, UK). Samples were stirred with a magnetic bar and flushed with He before injection of 1 mL sample aliquots (repeated 3 times). The DOC was then converted into gaseous $CO_2$ by combustion at 850 °C, quantitatively oxidized by copper oxide and separated from other combustion products in a reduction column and a water condenser. This $CO_2$ was transferred to the IRMS via an open split device. To avoid a significant memory effect between consecutive analyses, each sample (injected and measured three times) was separated by six injections of deionized water and the first sample measurement was discarded. Average $\delta^{13}C_{DOC}$ reproducibility was 1.0 ‰ for standards and 0.5 ‰ for samples (1SD). The average reproducibility for sample [DOC] measurements was 0.3 mM, and blank tests were below the detection limit.

In addition to DOC measurements, we calculated the "Total carbon concentration" as the sum of DOC, DIC, and POC concentrations, with DIC and POC data from Havas et al. (2023). The corresponding isotopic composition ($\delta^{13}C_{Total}$) was calculated as the weighted average of the three $\delta^{13}C$. The DIC and POC isotope data were also used to calculate isotopic differences with $\delta^{13}C_{DOC}$, expressed in the $\Delta^{13}C$ notation. The values for $\delta^{13}C_{DIC}$ and $\delta^{13}C_{POC}$ are detailed in Havas et al. (2023) and summarized in the results section.

## 4. RESULTS

The water columns of the four lakes were clearly stratified in May 2019 (Fig. 2; Havas et al., 2023). The epi-, meta-, and hypo-limnion layers of each lake were identified based on the thermocline depths, and correspond to the oxygen-rich, intermediate, and oxygen-poor layers in the four lakes, although the oxycline in La Preciosa is slightly thinner than the thermocline (~5 vs. 8 m). In the following, DIC, POC, $O_2$, chlorophyll a (Chl a), $NH_4$, P and $CO_{2(aq)}$ data are also presented.

### 4.1. Lake La Alberca de los Espinos

Bulk DOC had a concentration of ~ 0.4 mM throughout the water column, except at 7 and 17 m, where it peaked at 1.0 and 1.7 mM, respectively (Fig. 3). Its isotopic composition ($\delta^{13}C_{DOC}$) was comprised between -27.2 and -

25.1 ‰ except at 7 m, where it reached -14.7 ‰ (Fig. 3). It represented ~ 8% of total carbon on average, and 93% of the organic carbon present in the water column. Total C concentration increased downward from about 7 to 9 mM. The $\delta^{13}C_{total}$ decreased from -3.9 to -7.9 ‰ between 5 and 17 m and then increased to -3.2 ‰ at 25 m (Table 1). The isotopic difference between DOC and DIC ($\Delta^{13}C_{DOC-DIC}$) was between -21.2 and -25.2 ‰, except at 7 m depth, where it peaked to -12.4 ‰ (Fig. 4; Table 2). The $\Delta^{13}C_{DOC-POC}$ values were comprised between -1.5 and +3.1 ‰, except at 7 m depth, where DOC was enriched in $^{13}C$ by ~11.5 ‰ (Fig. 4; Table 2). The DIC concentration and $\delta^{13}C_{DIC}$ averaged 7.5 ± 0.7 mM and -2.9 ± 0.8 ‰; POC concentration and $\delta^{13}C_{POC}$ averaged 0.04 ± 0.02 mM and -27.1 ± 1.3 ‰. Dissolved oxygen showed a stratified profile with an oxycline layer transitioning from $O_2$-saturated to $O_2$-depleted conditions between 5 and 12 m depths (Fig. 2). Chl a concentration showed three distinct peaks at ~7.5, 12.5 and 17.5 m depths, all reaching ~4 µg/L (Fig. 2). The average $NH_4^+$ and P concentrations were 3.9 and 11.3 µM, respectively. The activity of $CO_{2(aq)}$ was $10^{-5.00}$ at 7 m depth and increased to $10^{-3.40}$ at the bottom of the lake.

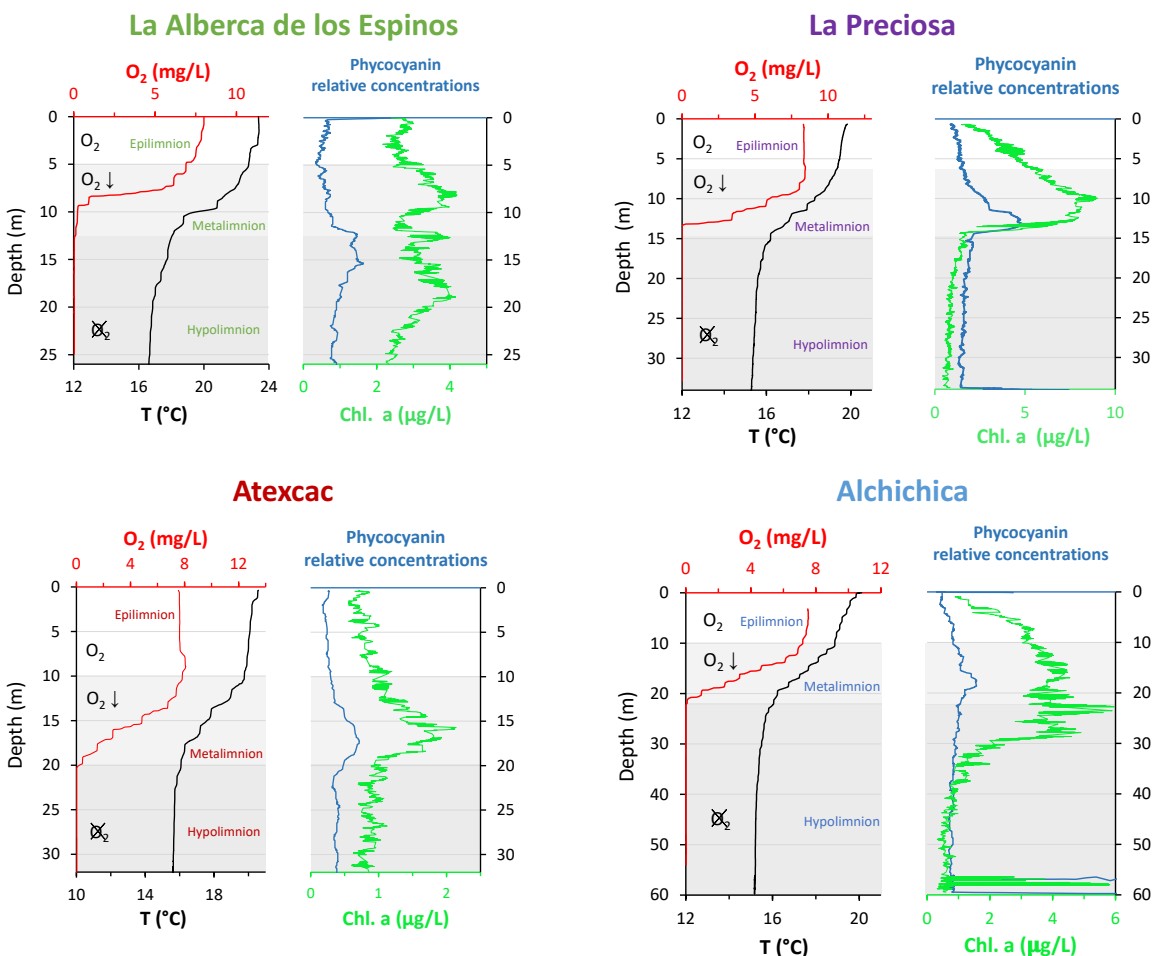

Figure 2. Physico-chemical parameter depth profiles of La Alberca de los Espinos, La Preciosa, Atexcac, and Alchichica: dissolved oxygen concentration (mg/L), water temperature (°C), phycocyanin and chlorophyll a pigments (µg/L). Absolute values for phycocyanin concentrations were not determined; only relative variations are represented (with increasing concentrations to the right). Epi-, meta- and hypo-limnion layers are represented for each lake by the white, gray, and dark gray areas, based on temperature profiles with the metalimnion corresponding to the thermocline. The three layers match the oxygen-rich, intermediate, and oxygen-poor zones, except in La Preciosa). Original data from Havas et al. (2023).

### 4.2. Lake La Preciosa

Bulk DOC had a concentration of ~ 0.5 mM throughout the water column except at 12.5 m, where it peaked at 1.6 mM. The $\delta^{13}C_{DOC}$ was -25.9 ± 0.4 ‰ throughout the water column except between 12.5 and 15 m, where it reached -20.0 ‰ (Fig. 3). The DOC represented ~ 3% of the total carbon on average, and 91% of the organic carbon present in the water column. The total C concentration was relatively stable at ~13.8 mM, while $\delta^{13}C_{total}$ was centered around -1 ‰ with a decrease to -2.8 ‰ at 12.5 m (Table 1). The $\Delta^{13}C_{DOC-DIC}$ values were very stable with depth around -26 ‰, but markedly increased at 12.5 m up to -19.8 ‰. (Fig. 4; Table 2). The $\Delta^{13}C_{DOC-POC}$ values decreased from ~1.3 ‰ in the upper waters to ~ -0.4 ‰ in the bottom waters but showed a peak to +7.1 ‰ at a depth of 12.5 m (Fig. 4; Table 2). The DIC concentration and $\delta^{13}C_{DIC}$ averaged 13.0 ± 0.8 mM and -0.2 ± 0.3 ‰; POC concentration and $\delta^{13}C_{POC}$ averaged 0.05 ± 0.02 mM and -26.1 ± 1.4 ‰. Dissolved oxygen showed a stratified profile with an oxycline layer transitioning from $O_2$-saturated to $O_2$-depleted conditions between 8 and 14 m depths (Fig. 2). The Chl a concentration showed a large peak at ~10 m, reaching 9 µg/L (Fig. 2). The average $NH_4^+$ and P concentrations were 1.9 and 0.2 µM, respectively. The activity of $CO_{2(aq)}$ averaged $10^{-4.57}$.

### 4.3. Lake Atexcac

Bulk DOC had a concentration of ~ 1.1 mM throughout the water column except at 16 and 23 m, where it reached 7.7 and 20.8 mM, respectively. The $\delta^{13}C_{DOC}$ increased from -20.0 to -8.7 ‰ between 5 and 23 m, decreasing to -11.2 ‰ at 30 m. It represented about 16% of the total carbon on average, and 98 % of the organic carbon present in the water column. Total C concentrations and $\delta^{13}C_{total}$ are centered around 27.7 mM and -0.6 ‰ with a clear increase to 38.9 mM and decrease to -2.7 ‰ at 23 m, respectively. The $\Delta^{13}C_{DOC-DIC}$ values significantly increased from the surface (-20.4 ‰) to the hypolimnion (~ -11.4 ‰). The DOC isotope compositions were strictly and significantly less negative than POC (i.e. enriched in heavy $^{13}C$), with $\Delta^{13}C_{DOC-POC}$ reaching as much as +17.9 ‰ at the depth of 23 m (Fig. 4; Table 2). The DIC concentration and $\delta^{13}C_{DIC}$ averaged 25.7 ± 0.9 mM and 0.5 ± 0.3 ‰; POC concentration and $\delta^{13}C_{POC}$ averaged 0.04 ± 0.02 mM and -27.7 ± 1.1 ‰. Dissolved oxygen showed a stratified profile with an oxycline layer transitioning from $O_2$-saturated to $O_2$-depleted conditions between 10 and 20 m depths (Fig. 2). Chl a concentration showed a small peak at 16 m, reaching 2 µg/L (Fig. 2). The average $NH_4^+$ and P concentrations were 2.5 and 0.3 µM, respectively. The activity of $CO_{2(aq)}$ averaged $10^{-4.27}$.

### 4.4. Lake Alchichica

Bulk DOC had a concentration of ~ 0.5 mM throughout the water column, except in the hypolimnion, where it reached up to 5.4 mM. The $\delta^{13}C_{DOC}$ varied from -29.3 to -25.1 ‰, with maximum values found in the hypolimnion (Fig. 3). The DOC represented about 5 % of the total carbon on average, and 93 % of the organic carbon present in the water column. Total carbon concentration depth profile roughly followed that of DOC, while $\delta^{13}C_{total}$ was between -0.2 and 1.6 ‰ throughout the water column, except in the lower part of the hypolimnion, where it decreased to -2.3 ‰ (Table 1). The isotopic difference between DOC and DIC ($\Delta^{13}C_{DOC-DIC}$) was slightly smaller in the hypolimnion and was comprised between -26.7 and -30.9 ‰. The DOC isotope compositions were more negative than POC, with $\Delta^{13}C_{DOC-POC}$ values between -0.7 and -3.5 ‰ (Fig. 4; Table 2). The DIC concentration and $\delta^{13}C_{DIC}$ averaged 34.6 ± 0.6 mM and 1.7 ± 0.2 ‰; POC concentration and $\delta^{13}C_{POC}$ averaged 0.01 ± 0.04 and -

25.6 ± 1.0 ‰. Dissolved oxygen showed a stratified profile with an oxycline layer transitioning from $O_2$-saturated to $O_2$-depleted conditions between ~10 and 20 m depths (Fig. 2). Chl a showed a broad peak between ~ 10 and 30 m, averaging 4 µg/L and with a narrow maximum of 6 µg/L (Fig. 2). The average $NH_4^+$ and P concentrations were 4.3 and 1.5 µM, respectively. The activity of $CO_{2(aq)}$ averaged $10^{-4.53}$.

Table 1
Concentration and isotopic composition of dissolved organic carbon (DOC). Total carbon concentration is the sum of DOC, DIC, and POC reservoirs. For LP 8m, [DIC] was not measured, and the total carbon concentration was not calculated. The DIC and POC were determined by Havas et al. (2023). The $\delta^{13}C_{Total}$ is the weighted average of the three $\delta^{13}C$. ND: non-determined.

| Lake | Sample | DOC | Total Carbon | $\delta^{13}C_{DOC}$ | $\delta^{13}C_{Total}$ |
|---|---|---|---|---|---|
| | | mmoles/L | | ‰ | |
| La Alberca de Los Espinos | Albesp 5m | 0.4 | 7.2 | -26.7 | -3.9 |
| | Albesp 7m | 1.0 | 8.1 | -14.7 | -3.9 |
| | Albesp 10m | 0.4 | 7.6 | -25.2 | -5.1 |
| | Albesp 17m | 1.7 | 9.0 | -26.3 | -7.9 |
| | Albesp 20m | 0.4 | 8.4 | -25.1 | -4.5 |
| | Albesp 25m | 0.4 | 9.2 | -27.2 | -3.2 |
| La Preciosa | LP 5m | 0.5 | 14.0 | -25.4 | -0.9 |
| | LP 8m | 0.9 | | ND. | ND. |
| | LP 10m | 0.3 | 13.7 | -25.7 | -0.4 |
| | LP 12.5m | 1.6 | 13.2 | -20.0 | -2.8 |
| | LP 15m | 0.5 | 13.9 | -24.0 | -1.3 |
| | LP 20m | 0.3 | 13.6 | -26.2 | -1.0 |
| | LP 31m | 0.3 | 13.6 | -26.2 | -0.9 |
| Atexcac | ATX 5m | 0.92 | 27.4 | -20.0 | -0.4 |
| | ATX 10m | 1.8 | 28.1 | -15.5 | -0.7 |
| | ATX 16m | 7.8 | 34.7 | ND. | ND. |
| | ATX 23m | 21.0 | 45.2 | -8.7 | -3.6 |
| | ATX 30m | 0.7 | 26.4 | -11.2 | -0.1 |
| Alchichica | AL 5m | 0.7 | 35.8 | ND. | ND. |
| | AL 10m | 0.4 | 33.5 | -28.3 | 1.6 |
| | AL 20m | 0.4 | 35.0 | -29.3 | 1.3 |
| | AL 30m | 0.4 | 35.1 | -28.3 | 1.2 |
| | AL 35m | 2.3 | 37.2 | -26.8 | -0.2 |
| | AL 40m | 2.2 | 37.0 | -25.8 | -0.1 |
| | AL 50m | 5.0 | 39.8 | -25.1 | -1.8 |
| | AL 55m | 0.5 | 35.3 | -27.6 | 1.1 |
| | AL 58m | 5.4 | 40.2 | -27.7 | -2.3 |
| | AL 60m | 0.7 | 35.3 | -26.1 | 1.0 |

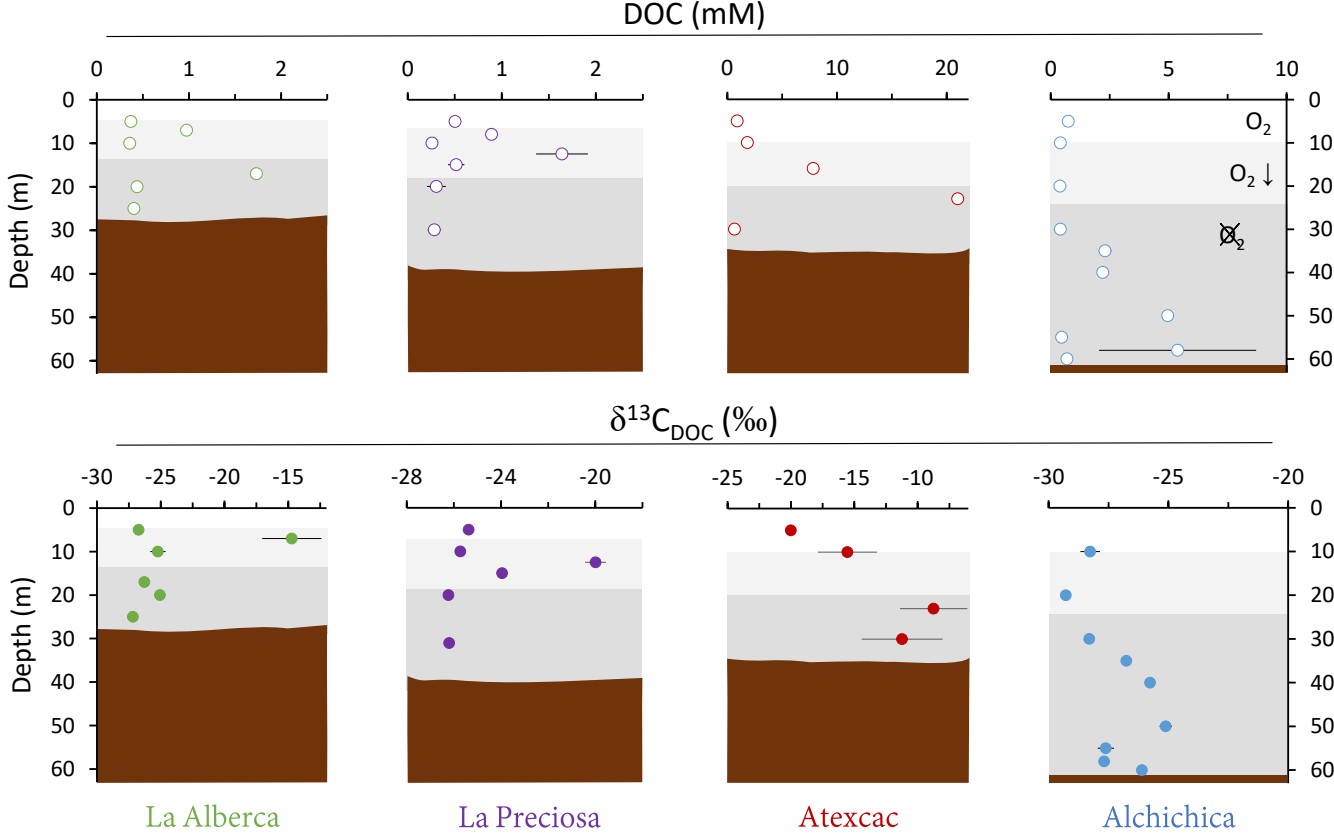

256

Figure 3. Vertical profiles of concentration and isotopic composition of dissolved organic carbon (DOC) throughout the water columns of the studied lakes: La Alberca de los Espinos, La Preciosa, Atexcac, and Alchichica. Concentration is in mmol/L (mM) and isotopic composition in ‰ vs. VPDB. The white, gray, and dark gray shading is as in Fig. 2. The brown shading symbolizes the presence of sediment at the bottom of the water columns (showing the greater water depth in Lake Alchichica).

262

263

264

265

## 5. DISCUSSION

The four Mexican lakes studied here have a high DOC content but very different profiles and signatures for [DOC] and $\delta^{13}C_{DOC}$ (Fig. 3). Evaporation may increase DOC concentration (Anderson and Stedmon, 2007; Zeyen et al., 2021), but would not explain the significant intra-lake DOC variability with depth. It is likely marginal because, in contrast with what was observed for DIC (Havas et al., 2023), there is no correlation between the average DOC concentration in the Mexican lakes and their salinity ($R^2=0.47$, $p=0.2$ for DOC and $R^2=0.93$, $p=5.8*10^{-5}$ for DIC). In the following discussion, we therefore explore the different patterns of DOC production and fate, in relation to other environmental and biological variations, and how this can provide information about past DOC-related perturbations of the C cycle.

276

## 5.1 Sources and fate of DOC

Due to their endorheic nature, the four lakes receive relatively little allochthonous OM (Alcocer et al., 2014b; Havas et al., 2023). It is therefore possible to focus on DOC-related processes occurring within the water column, particularly on autochthonous DOC primary production. Autochthonous DOC can form through higher-rank OM degradation processes such as sloppy feeding by zooplankton, UV photolysis or bacterial and viral cell lysis (Lampert, 1978; Hessen, 1992; Bade et al., 2007; Thornton, 2014; Brailsford, 2019), as well as passive (leakage) or active (exudation) release by healthy cells (e.g. Baines and Pace, 1991; Hessen and Anderson, 2008; Thornton, 2014; Ivanovsky et al., 2020). Generally, this C release (whether "active" or "passive") tends to be enhanced in nutrient-limited conditions because recently fixed C is in excess compared with other essential nutrients such as N or P (Hessen and Anderson, 2008; Morana et al., 2014; Ivanovsky et al., 2020). For oxygenic phototrophs, this is particularly true under high photosynthesis rates, because photorespiration bolsters the excretion of DOC (Renstrom-Kellner and Bergman, 1989). Oligotrophic conditions also tend to limit heterotrophic bacterial activity and thus preserve DOC stocks (Thornton, 2014; Dittmar, 2015). Both these production and preservation aspects are consistent with the trend of increasing DOC concentrations observed in the lakes, from the less oligotrophic La Alberca and La Preciosa (0.7 mM on average) to the more oligotrophic Alchichica (1.8 mM) and Atexcac (6.5 mM).

### 5.1.1 DOC release by autotrophs

In the four Mexican lakes, DOC concentration profiles exhibit one or several peaks occurring in both oxic and anoxic waters (Fig. 3). In La Alberca and La Preciosa, these peaks correlate with Chl a peaks, but not in the other two lakes. However, in Atexcac, a remarkable DOC peak (Fig. 3) occurs at the same depth as a peak of anoxygenic photosynthesis (Havas et al., 2023). These co-occurrences indicate that a large portion of DOC in these three lakes (at least at these depths) arises from the release of photosynthetic C fixed in excess. Phytoplankton in aerobic conditions generally releases dissolved organic matter by (i) an active "overflow mechanism" (DOM exudation) or (ii) passive diffusion through the cell membranes, but this remains to be shown for anoxygenic organisms. In the first case, DOM is actively released from the cells as a result of C fixation rates higher than growth and molecular synthesis rates (e.g. Baines and Pace, 1991). Hence, DOM exudation depends not only on the nature of primary producers (different taxa may display very different growth rates, photosynthetic efficiency, and exudation mechanisms), but also on environmental factors such as irradiance and nutrient availability (e.g. Otero and Vincenzini, 2003; Morana et al., 2014; Rao et al., 2021). Exudation of DOM may also serve "fitness-promoting purposes" such as storage, defense, or mutualistic goals (Bateson and Ward, 1988; Hessen and Anderson, 2008). In the case of passive diffusion, DOM release also depends on cell permeability and the outward DOC gradient, but is more directly related to the amount of phytoplankton biomass (e.g. Marañón et al., 2004). Thus, any new photosynthate production drives a steady DOM release rate, independent of environmental conditions to some extent (Marañón et al., 2004; Morana et al., 2014). The fact that La Alberca and La Preciosa have lower DOC but Chl a concentrations higher than Atexcac and Alchichica overall, suggests that DOC production does not directly relate to phytoplankton biomass and is not passively released. By contrast, active DOC release is supported by DOC isotope signatures. These tropical Mexican lakes correspond precisely to environmental contexts (high

irradiance and oligotrophic freshwater bodies) where DOC exudation has been observed and is predicted (e.g. Baines and Pace, 1991; Morana et al., 2014; Thornton, 2014; Rao et al., 2021).

Release of DOC by primary producers can be characterized by the percentage of extracellular release (PER), which corresponds to the fraction of DOC over total (dissolved and particulate) OM primary production (e.g. Thornton et al., 2014). The PER is highly variable and averages about 13% of C biomass over a wide range of environments (e.g. Baines and Pace, 1991; Thornton, 2014). Values as high as 99% have been reported (see Bertilsson and Jones, 2003), showing that most of the fixed C can be released in the external aqueous media as DOC. At depths where oxygenic photosynthesis occurs, the DOC over total OC ratio averages approximately 95, 94, 99, and 85 % for La Alberca, La Preciosa, Atexcac, and Alchichica, respectively. Thus, although the PER was not directly measured, and some of the measured DOC may correspond to an older long-term DOC reservoir, the majority of DOC measured could represent a recent phytoplankton exudation.

The DOC peaks associated with primary production (mainly photosynthesis) are characterized by very positive $\Delta^{13}C_{DOC-POC}$ (from +3 to +18 ‰, Fig. 4). These signatures further support a primary origin of DOC as photosynthate release at these depths, rather than a secondary origin by OM degradation. Bacterial heterotrophy would generate smaller and rather negative $\Delta^{13}C_{DOC-POC}$ (section 5.1.2. and references therein) and cell lysis or zooplankton sloppy feeding would also produce $\delta^{13}C_{DOC}$ close to $\delta^{13}C_{POC}$ values. Photo-degradation is unlikely to proceed at these depths and would not generate such positive fractionations (Chomicki, 2009). A switch from $CO_{2(aq)}$ to $HCO_3^-$ as an inorganic C source (which differ by 10‰, e.g. Mook et al., 1974) would not adequately explain the deviation between $\delta^{13}C_{POC}$ and $\delta^{13}C_{DOC}$. The isotopic enrichment of DOC molecules relative to POC must therefore have a different origin.

The $^{13}$C-enriched DOC could originate from photosynthetic organisms using a different C-fixation pathway, inducing a smaller isotopic fractionation (provided that these organisms contributed predominantly to the DOC rather than to the POC fraction). In La Alberca and Atexcac, anoxygenic phototrophic bacteria may release large amounts of DOC, especially under nutrient-limiting conditions (Ivanovsky et al., 2020). Unlike cyanobacteria or purple sulfur bacteria (PSB, anoxygenic phototrophs belonging to the Proteobacteria), which use the Calvin-Benson-Bassham pathway (CBB), green sulfur bacteria (GSB; another group of anoxygenic phototrophs belonging to the Chlorobi), use the reductive citric acid cycle or reverse tricarboxylic-TCA cycle, which tends to induce smaller isotopic fractionations (between ~ 3–13 ‰, Hayes, 2001). The DOC isotope signatures recorded in the hypolimnion of La Alberca ($\varepsilon_{DOC-CO2} \approx$ -13.5 ± 2 ‰) agree well with fractionations found for this type of organism in laboratory cultures and in stratified water bodies (Posth et al., 2017). By contrast, $\varepsilon_{DOC-CO2}$ signatures in the hypolimnion of Atexcac are higher ($\varepsilon_{DOC-CO2} \approx$ 0 ‰), and thus cannot be explained by the use of the reductive citric acid cycle C fixation pathway. Consistently, GSB were identified in La Alberca but not in Atexcac (Havas et al., 2023).

Phytoplankton blooms may specifically release isotopically heavy organic molecules. Carbohydrates could be preferentially released under nutrient-limiting conditions as they are devoid of N and P (Bertilsson and Jones, 2003; Wetz and Wheeler, 2007; Thornton, 2014). Carbohydrates typically have a $^{13}$C-enriched (heavy) isotopic composition (Blair et al., 1985; Jiao et al., 2010; Close and Henderson, 2020). Considering the isotopic mass

balance of cell specific organic compounds, this molecular hypothesis is insufficient to explain the full range of
$\Delta^{13}C_{DOC-POC}$ variations measured in La Alberca and Atexcac (Hayes, 2001).
Alternatively, such enrichments require that DOC and DIC first accumulate in the cells. If DOC molecules were
released as soon as they were produced, their isotopic composition would tend towards that of the biomass (i.e.
$\delta^{13}C_{POC}$, within the range of molecule-specific isotopic compositions), which is not the case. If DIC could freely
exchange between inner and outer cell media, maximum "carboxylation-limited" fractionation (between ~ 18 and
30 ‰ depending on RuBisCO form, Thomas et al., 2019) would be expressed in all synthesized organic molecules,
as represented in Fig. 5a (e.g. O'Leary, 1988; Descolas-Gros and Fontungne, 1990; Fry, 1996). This is also
inconsistent with the DOC isotopic signatures (see $\varepsilon_{DOC-CO2}$ in Table. 2).
Under the environmental conditions of the lakes studied (i.e. low $CO_2$ relative to $HCO_3^-$, local planktonic
competition for $CO_2$, and low nutrient availability), the activation of an intracellular DIC-concentrating mechanism
(DIC-CM) is expected (Beardall et al., 1982; Burns and Beardall, 1987; Fogel and Cifuentes, 1993; Badger et al.,
1998; Iñiguez et al., 2020). This mechanism is particularly relevant in oligotrophic aqueous media (Beardall et al.,
1982), where $CO_2$ diffusion is slower than in the air (O'Leary, 1988; Fogel and Cifuentes, 1993; Iñiguez et al.,
2020). A DIC-CM has been proposed to reduce the efflux of DIC from the cells back to the extracellular solution.
This internal DIC is eventually converted into organic biomass, thereby drawing the cell isotopic composition
closer to that of $\delta^{13}C_{DIC}$ (Fig. 5; Beardall et al., 1982; Fogel and Cifuentes, 1993; Werne and Hollander, 2004). As
a conceptual model, we suggest that the activation of a DIC-CM could preserve a large $\Delta^{13}C_{POC-DIC}$, while
generating an apparent fractionation between the DOC and POC molecules. The initially fixed OC would be
discriminated against the heavy C isotopes, and incorporated into the cellular biomass (Fig. 5c, '$t_i$'). In turn,
following the overflow mechanism scenario, high photosynthetic rates (due to high irradiance and temperature,
and high DIC despite low $CO_2$) coupled with low population growth rates and organic molecule synthesis (due to
limited abundances of P, N, and Fe), would result in the exudation of excess organic molecules with heavy $\delta^{13}C_{DOC}$,
as they are synthesized from residual internal DIC, which progressively becomes $^{13}$C-enriched (Fig. 5c, '$t_{ii}$'). This
process could explain the formation of DOC with $\delta^{13}$C very close to DIC/$CO_2$ signatures as observed in Lake
Atexcac. This suggests that oligotrophic conditions could be a determinant factor in the generation of significantly
heavy $\delta^{13}C_{DOC}$, even more so if they are coupled to high irradiance. This also demonstrates that considerable
isotopic variability can exist between these two organic C reservoirs.
In summary, the unusual [DOC] and $\delta^{13}C_{DOC}$ profiles in La Alberca, La Preciosa and Atexcac could be interpreted
as mainly reflecting a prominent exudation of autochthonous C, fixed in excess by oxygenic and/or anoxygenic
phototrophs in nutrient-poor and high-irradiance conditions. The striking $^{13}$C-rich signatures of these exudates are
interpreted as reflecting either the activation of a DIC-CM by oxygenic and/or anoxygenic phototrophs or the
fixation of C *via* the reductive citric acid cycle. We propose a conceptual model involving the DIC-CM, whereby
oligotrophic and high irradiance contexts can lead to high $\delta^{13}C_{DOC}$ compared to both $\delta^{13}C_{DIC}$ and $\delta^{13}C_{POC}$.

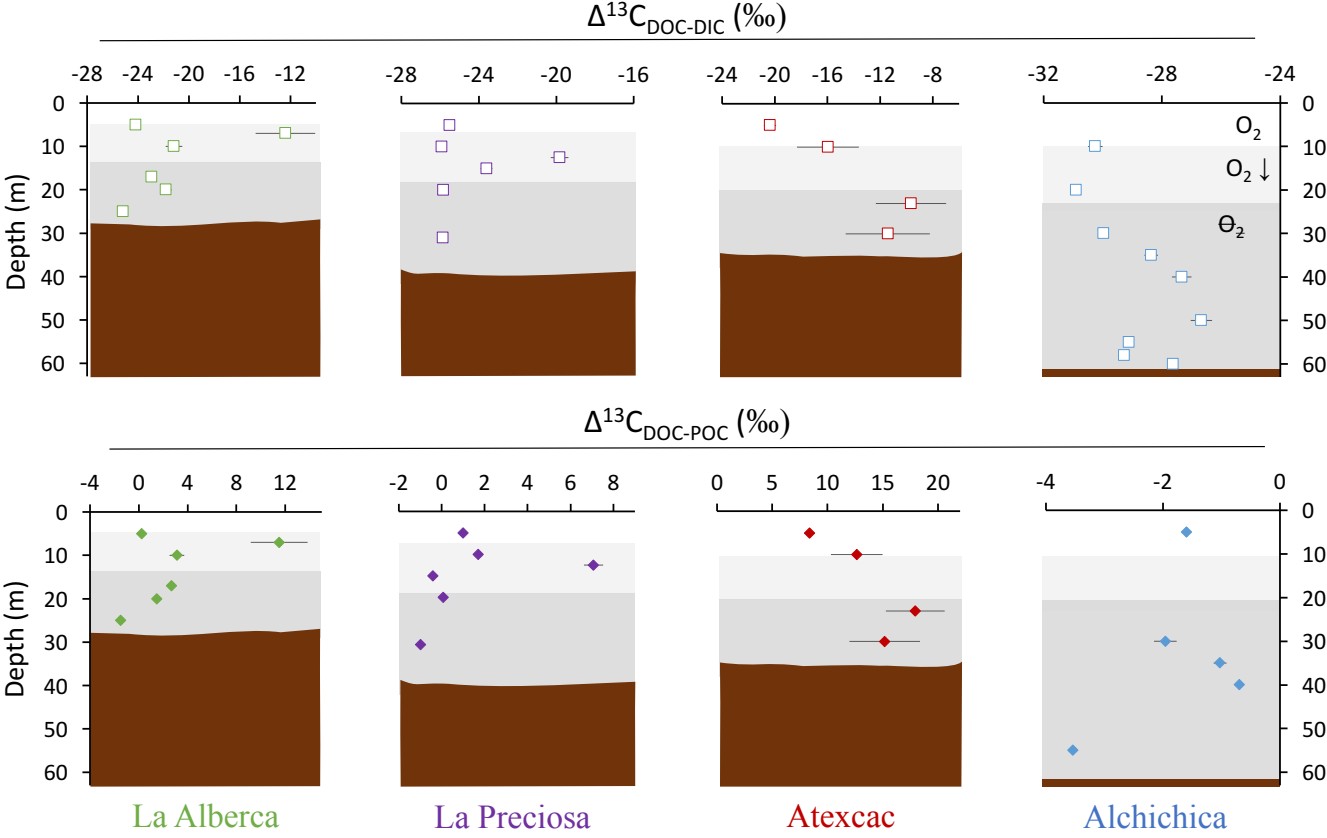


Figure 4. Vertical profiles of the difference in $\delta^{13}$C values of DOC and DIC (top) as well as DOC and POC (bottom) throughout the water columns of the four lakes (all expressed as $\Delta^{13}$C in ‰ *vs.* VPDB). POC and DIC data used in these calculations are from Havas et al. (2023). In Alchichica, $\delta^{13}C_{DOC}$ was not measured at 5 m and its value at 10 m was used in this calculation of $\Delta^{13}C_{DOC-POC.}$ The white, gray, and dark gray shading is as in Fig. 2. The brown shading symbolizes the presence of sediment at the bottom of the water columns.















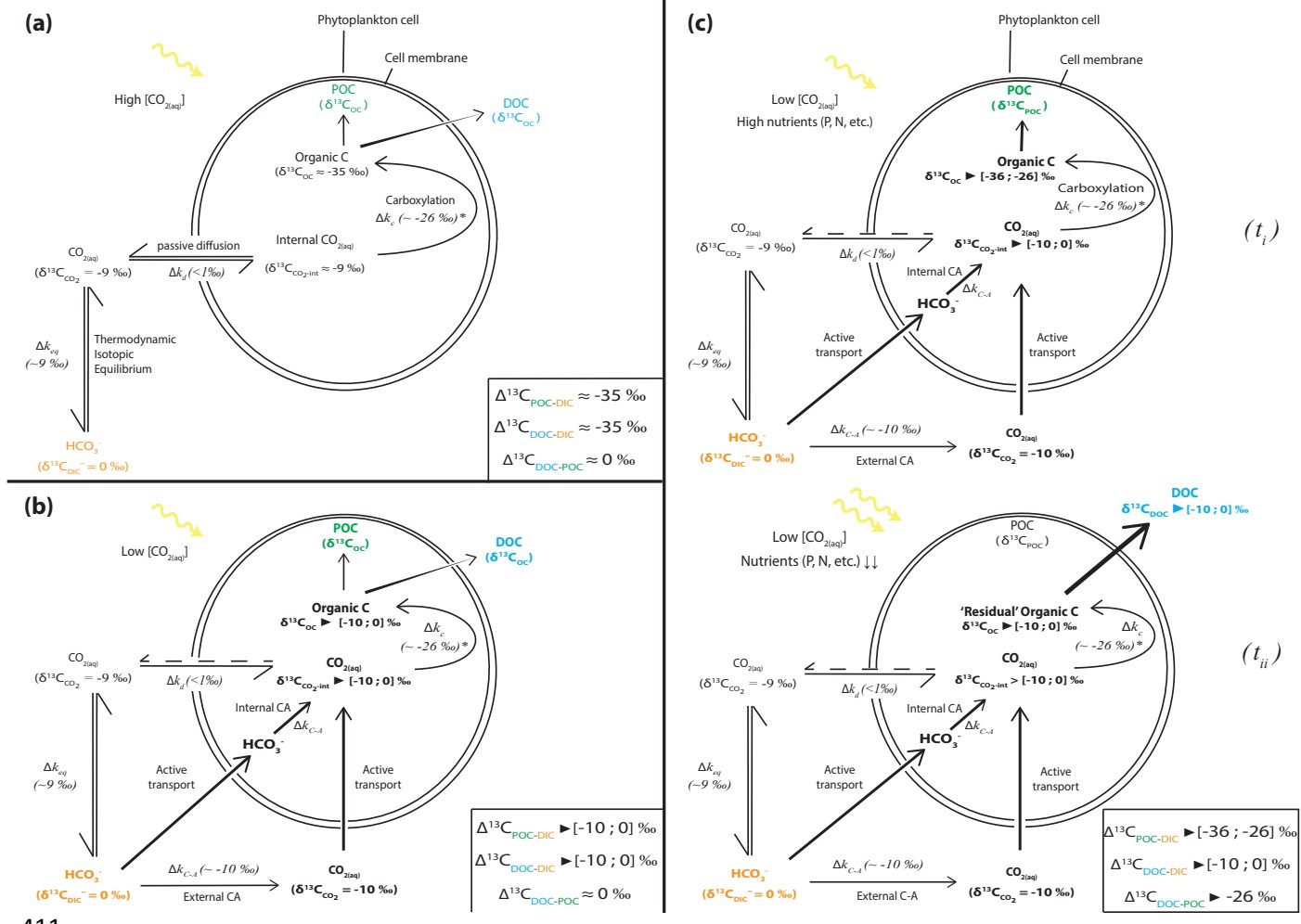


Figure 5. Schematic view of phytoplankton cells during autotrophic C fixation through different C supply
strategies and associated apparent isotopic fractionation between DIC and POC/DOC and between DOC and POC.
(a) Case where [$CO_{2(aq)}$] is high enough to allow for a DIC supply by passive $CO_{2(aq)}$ diffusion through the cell
membrane and $CO_{2(aq)}$ is at equilibrium with other DIC species. Isotopic fractionation is maximum (minimum
$\delta^{13}C_{OC}$) because C fixation is limited by the carboxylation step. DOC is released following an in- to outward cell
concentration gradient and has a similar composition to POC. (b) "Classic" view of C isotopic cycling resulting
from active DIC transport within the cell because of low ambient [$CO_{2(aq)}$] (through a DIC-CM). Carbonic anhydrase
(CA) catalyzes the conversion between $HCO_3$ and $CO_{2(aq)}$ inside or outside the cell with isotopic fractionation close
to equilibrium fractionation (~ 10 ‰). While inward passive $CO_{2(aq)}$ diffusion can still occur, the DIC-CM activation
reduces the reverse diffusion, resulting in internal $CO_{2(aq)}$ isotopic composition approaching that of the incoming
DIC (depending on the fraction of internal $CO_{2(aq)}$ leaving the cell). Acting as a "closed-system", most of the
internal DIC is fixed as OC, and minimum isotopic fractionation is expressed for both POC and DOC. (c) Proposed
model for C isotopic fractionation with active DIC transport including isotopic discrimination between POC and
DOC. ($t_i$) Initially fixed C is isotopically depleted and incorporates the cell's biomass as long as there are sufficient
nutrients to enable the synthesis of "complex" organic molecules. ($t_{ii}$) In low nutrient conditions, but with high
photosynthetic activity – subsequently fixed C is released out of the cell as DOC following the "overflow"
hypothesis and inherits heavier isotopic compositions from the residual internal DIC. This leads to distinct POC
and DOC isotopic signatures, with small fractionation between DOC and DIC, the amplitude of which will depend
on the rate of $CO_2$ backward diffusion, and the biomass C (POC) to released C (DOC) ratio.

**5.1.2 OM partial degradation and DOC accumulation: the case of Lake Alchichica**

From the previous discussion, it appears that the environmental conditions of the Mexican lakes favor substantial phytoplankton production of DOC. Alcocer et al. (2014a) proposed that an early spring cyanobacterial bloom in Lake Alchichica may favor the production of DOC in the epilimnion. However, at the time of sampling, the DOC reservoir in this lake was not correlated with any sizeable autotrophic activity at any depth. The large epilimnetic Chl a peak did not correlate with any changes in [DOC] or $\delta^{13}C_{DOC}$ (Fig. 3). Compared with the other lakes, the geochemical conditions in which Chl a was produced in Alchichica may have been incompatible with the activation of a DIC-CM and significant DOC exudation. Alchichica had similar $[CO_{2(aq)}]$ to La Preciosa, but higher P and $NH_4^+$ concentrations (Havas et al., 2023); La Alberca had higher P concentrations, but similar $[NH_4^+]$ and lower $[CO_{2(aq)}]$. In contrast with measurements from 2013 (Alcocer et al., 2014a), we found a large increase in DOC in the middle of the anoxic hypolimnion of Alchichica, which did not correspond to any change in the DIC reservoir, unlike that observed for La Preciosa at 12.5 m and Atexcac at 23 m (Havas et al., 2023). At these depths, photosynthetic active radiation (PAR) is below 0.1% in Alchichica during the stratified season (Macek et al., 2020), which might not be sufficient to trigger major anoxygenic phytoplankton DOC release.

The DOC reservoir in Alchichica is characterized by a $\delta^{13}C_{DOC}$ (and $\Delta^{13}C_{DOC-DIC}$) lower than in the other lakes, systematically showing $^{13}$C-depleted signatures relative to POC (i.e. $\delta^{13}C_{DOC} < \delta^{13}C_{POC}$; Fig. 4). Thus, if the DOC increase in the hypolimnion of Alchichica resulted from the release of photosynthetic OC, as in some of the other lakes, it was not associated with the same C isotope fractionation (e.g. if anoxygenic phototrophs did not concentrate intracellular DIC, cf. Fig. 5a). Some PSB have been identified but they only become abundant toward the end of the stratification (from July/August to December/January; Alcántara-Hernández et al., 2022; Iniesto et al., 2022).

Alternatively, the hypolimnetic DOC increase in Lake Alchichica may reflect the preservation and accumulation of DOM over the years, consistent with the higher [DOC] measured in 2019 than in the previous years (Alcocer et al., 2014a). While alteration of the DOC reservoir by UV-photolysis would induce positive isotopic fractionation (Chomicki, 2009), the slightly negative $\Delta^{13}C_{DOC-POC}$ signatures support the possibility of DOC being mainly a recalcitrant residue of primary OM degradation by heterotrophic organisms (Alcocer et al., 2014a). The preferential consumption of labile $^{13}$C-enriched molecules by heterotrophic bacteria would leave the residual OM with more negative isotopic signatures (Williams and Gordon, 1970; Lehmann et al., 2002; Close and Henderson, 2020). The DIC and POM data were also consistent with heterotrophic activity from the surface to the hypolimnion of Alchichica, by recording complementary decreasing and increasing $\delta^{13}C$, respectively, and a decreasing C:N ratio (Havas et al., 2023). Degradation by heterotrophic bacteria leaves more recalcitrant DOM in the water column, which tends to accumulate over longer periods of time (Ogawa et al., 2001; Jiao et al., 2010; Kawasaki et al., 2013). The DOM content is a balance between production by autotrophs and consumption by heterotrophs, especially in environments where both types of organisms compete for low-concentration nutrients (Dittmar, 2015). If the DOC in Alchichica represents a long-term reservoir, its presence might favor the development of bacterial populations. A shift of the cyanobacterial DOC from the epilimnion toward the hypolimnion of Alchichica was described at the end of the spring (Alcocer et al., 2014a; 2022). Thus, part of the hypolimnetic DOC in Alchichica may originate from a phytoplankton release, as observed in the other lakes, but it was already partially degraded by heterotrophic bacteria at the time we sampled it. The deeper and darker anoxic waters of

Alchichica could help to better preserve this DOC from intense microbial and light degradation, hence allowing
its accumulation over time.
In conclusion, the DOC reservoir in Alchichica (notably in the hypolimnion) more likely represents an older, more
evolved DOM pool. The time required for its accumulation and long-term stability has not yet been evaluated.



Table 2
Isotopic fractionation between DOC and DIC, and DOC and POC, where $\Delta^{13}C_{x-y} = \delta^{13}C_x - \delta^{13}C_y$ is the apparent
fractionation and $\varepsilon$ is computed as the actual metabolic isotopic discrimination between $CO_2$ and DOC. In
Alchichica, $\delta^{13}C_{DOC}$ was not measured at 5 m, and its value at 10 m was used in this calculation of $\Delta^{13}C_{DOC-POC}$. The
full chemistry at depths 35 and 58 m was not determined, thus the calculation of $\delta^{13}C_{CO_2}$ for these samples is
based on the composition of samples above and below. Isotopic data for DIC, POC, and $CO_2$ are from Havas et al.
484     (2023).

| Lake | Sample | $\Delta^{13}C_{DOC-DIC}$ | $\Delta^{13}C_{DOC-POC}$ | $\varepsilon_{DOC-CO2}$ |
|---|---|---|---|---|
| | | ‰ | | ‰ |
| La Alberca de Los Espinos | Albesp 5m | -24.2 | 0.2 | -14.8 |
| | Albesp 7m | -12.4 | 11.5 | -3.0 |
| | Albesp 10m | -21.2 | 3.1 | -11.6 |
| | Albesp 17m | -22.9 | 2.7 | -13.1 |
| | Albesp 20m | -21.8 | 1.5 | -12.2 |
| | Albesp 25m | -25.2 | -1.5 | -15.9 |
| La Preciosa | LP 5m | -25.5 | 1.0 | -15.7 |
| | LP 10m | -25.9 | 1.7 | -16.0 |
| | LP 12.5m | -19.8 | 7.1 | -9.8 |
| | LP 15m | -23.6 | -0.4 | -13.5 |
| | LP 20m | -25.8 | 0.1 | -15.7 |
| | LP 31m | -25.8 | -1.0 | -15.7 |
| Atexcac | ATX 5m | -20.4 | 8.4 | -10.6 |
| | ATX 10m | -16.0 | 12.6 | -6.1 |
| | ATX 23m | -9.7 | 17.9 | 0.6 |
| | ATX 30m | -11.4 | 15.2 | -1.2 |
| Alchichica | AL 5m | ND. | -1.6 | |
| | AL 10m | -30.3 | | -20.1 |
| | AL 20m | -30.9 | | -20.5 |
| | AL 30m | -30.0 | -2.0 | -19.5 |
| | AL 35m | -28.4 | -1.0 | -17.9 |
| | AL 40m | -27.3 | -0.7 | -16.8 |
| | AL 50m | -26.7 | | -16.2 |
| | AL 55m | -29.1 | -3.5 | -18.7 |
| | AL 58m | -29.3 | | -18.8 |
| | AL 60m | -27.6 | | -17.1 |



## 5.2 DOC analysis provides deeper insights into planktonic cell functioning and water column C cycle dynamics than POC or DIC analyses

The depth profiles of DOC concentration and isotope composition differ significantly from those of POC. Notably in La Preciosa, the photosynthetic DOC production (+1.5 mM) at the Chl a peak depth matches the decrease in DIC (- 2 mM), while there was no change in [POC] or $\delta^{13}C_{POC}$ (Havas et al., 2023). Just below, at 15 m depth, $\delta^{13}C_{POC}$ exhibited a marked increase (+3.6 ‰) interpreted as reflecting heterotrophic activity (Havas et al., 2023). It is likely explained by the production of DOC with heavier isotope compositions between 12.5 and 15 m depth, and its consumption by heterotrophic organisms (as seen with $\Delta^{13}C_{DOC-POC} \approx 0$). In La Alberca, the peaks of oxygenic and anoxygenic photosynthesis clearly stand out from DOC concentrations (+ 0.5/1.5 mM), but not from POC concentrations (+ <0.03 mM), while the DIC geochemical signatures reflected the influence of OC respiration, sediment-associated methanogenesis, and possible volcanic degassing at the bottom of the lake (Havas et al., 2023). In Atexcac, anoxygenic photosynthesis is clearly evidenced by [DOC] and $\delta^{13}C_{DOC}$ data (see 5.1.1), but is not recorded by the POC reservoir (a decrease of 0.03 mM at this depth) and not as distinctively by the DIC reservoir (a decrease of ~ 2 mM; Havas et al., 2023). It implies that recently fixed OC is quickly released out of the cells as DOC, transferring most C from DIC to DOC, rather than POC, which is therefore an incomplete archive of the biogeochemical reactions occurring in water columns. The isotopic analysis of DIC, and by extension of authigenic carbonates, especially in alkaline-buffered waters, might not be sensitive enough to faithfully record all environmental and biological changes.

The $\delta^{13}C_{DOC}$ recorded in La Alberca, La Preciosa, and Atexcac present peculiar heavy signatures, which provide strong constraints on planktons intra-cellular functioning and their use of C. These signatures may arise from the activation of a DIC-CM or from a specific metabolism or C-fixation pathway. By contrast, the use of a DIC-CM is poorly captured by $\delta^{13}C_{POC}$, although recognition of active DIC uptake has often been based on this signal (by reduced isotopic fractionation with DIC; e.g. Beardall et al., 1982; Erez et al., 1998; Riebesell et al., 2000). Most interestingly, intra-cellular amorphous Ca-carbonates (iACC) are formed in some of the cyanobacteria from Alchichica microbialites, possibly due to supersaturated intra-cell media following active DIC uptake through a DIC-CM (Couradeau et al., 2012; Benzerara et al., 2014). While the link between DIC-CM and iACC still needs to be demonstrated (Benzerara et al., 2014), the active use of DIC-CMs in Mexican lakes is independently supported by the DOC isotopic signature.

In summary, the analysis of DOC concentrations and isotope compositions showed that most of the autochthonous C fixation ends up in the DOC reservoir, thus highlighting important features of the lakes and their C cycle that were not evidenced by POC and DIC analyses alone, notably the activation of a DIC-CM and a better description of the planktonic diversity. In the future, it will be interesting to couple the present analyses with deeper molecular and compound-specific isotopic analyses of DOM (Wagner et al., 2020).

## 5.3 Implications for the hypothesis of a large DOC reservoir controlling past carbon cycling

In these Mexican lakes, the DOC concentrations (from 0.6 to 6.5 mM on average) are between 14 and 160 times higher than the POC concentrations. The DOC represents from 5 to 16% of the total C measured in the four lakes. In comparison, it remains under 0.3 mM in large-scale anoxic basins such as the Black Sea (Ducklow et al., 2007).

In the modern ocean, DOC is also the main organic pool but its concentration rarely exceeds 0.08 mM (Hansell,
2013). Thus, the DOC pools of these lakes is much larger than in the modern ocean and can be used to draw
comparisons with studies invoking past occurrences of oceanic carbon cycles dominated by vast DOC reservoirs
(e.g. Rothman et al., 2003; Sexton et al., 2011).

**5.3.1 Eocene carbon isotope excursions (CIEs)**
Ventilation/oxidation cycles of a large deep ocean DOC reservoir have been inferred to explain carbonate isotopic
records of successive warming events through the Eocene (Sexton et al., 2011). In this scenario, the release of
carbon dioxide into the ocean/atmosphere system following DOC oxidation would trigger both the precipitation
of low $\delta^{13}$C carbonates and an increase of the atmospheric greenhouse gas content. The size of this DOC reservoir
should have been at least 1600 PgC (about twice the size of the modern ocean DOC reservoir) to account for a 2–
4°C increase in deep ocean temperatures (Sexton et al., 2011). However, the main counter-argument to this
hypothesis is that the buildup of such a DOC reservoir at modern DOC production rates implies sustained deep
ocean anoxia over several hundred thousand years, while independent geochemical proxies do not support the
persistence of such anoxic conditions (Rigwell and Arndt, 2015). Our study suggests, albeit at a different scale,
that this kinetic argument may be weak. In these Mexican lakes, the lowest recorded [DOC] is 260 µM (Table 1),
which is about 6 times the deep modern ocean concentration (~ 45 µM; Hansell, 2013). Yet the entire water
columns of these lakes down to the surficial sediments are seasonally mixed with di-oxygen, showing that high
[DOC] (notably in Alchichica, which likely harbors a "long-term" DOC reservoir) can be achieved despite frequent
oxidative conditions. The oxidation of only half of the DOC in the lakes would generate average $\delta^{13}C_{DIC}$ deviations
between -0.6 and -1 ‰, corresponding to the C isotope excursion magnitudes described by Sexton et al. (2011).
Similarly, deep anoxic waters in the Black Sea hold about 3 times the amount of DOC found in the modern deep
open ocean (Ducklow et al., 2007; Sexton et al., 2011; Dittmar, 2015). In the Black Sea and in the Mexican lakes,
low nutrient availability may limit sulfate-reduction despite high sulfate and labile organic matter concentrations,
thus favoring DOM preservation and accumulation (Dittmar, 2015 and references therein). Margolin et al. (2016)
argued that substantial DOM is maintained in the Black Sea by large terrigenous inputs only. Our study attests the
possibility for "autochthonous systems" to reach DOC concentrations well above what is found in the Black Sea,
without requiring terrigenous inputs. Therefore, it supports the hypothesis that the buildup of a large DOC reservoir
may have influenced the carbonate isotopic record of Eocene warming events. Nonetheless, it remains to be proven
how this could apply to larger oceanic-type basins, with more variable environmental conditions (e.g. tropical vs.
polar latitudes), greater diversity of eukaryotic heterotrophs (in Phanerozoic oceans), and more active water
currents and ventilation processes. A better characterization of the molecular composition of DOM in the Mexican
lakes will help to understand how it can accumulate over time and refine the suggested analogy with Phanerozoic
CIEs. Furthermore, investigating the paleo-ecology and -geography of the CIE time period will also help to
constrain the potential applicability of a large DOC hypothesis (Sexton et al., 2011).



### 5.3.2 Neoproterozoic carbon isotope excursions (CIEs)

The presence of a large oceanic DOC reservoir has also been used to account for the Neoproterozoic C isotope record, where carbonates show $\delta^{13}C$ negative excursions of more than 10‰ over tens of Ma (Rothman et al., 2003; Fike et al., 2006; Swanson-Hysell et al., 2010; Tziperman et al., 2011). Once again, this hypothesis has been questioned because of (i) the oversized DOC reservoir (10 times the contemporaneous DIC, i.e., $10^2$ to $10^3$ times that of modern DOC) and (ii) the amount of oxidants required to generate such a sustained DOC oxidation process (see Ridgwell and Arndt, 2015). Recent studies offered potential explanations for this latter issue showing that pulses of continental weathering and an associated increase of sulfate supply would have provided sufficient oxidant (Shields et al., 2019; Chen et al., 2022), while lateral heterogeneity of the carbonate geochemical signatures – with a restricted record of the CIEs on the continental shelves – would require lower amounts of oxidant (Li et al., 2017; Shi et al., 2017).

Critically though, direct evidence for the existence of such high oceanic DOC levels in the past remains scarce (Li et al., 2017), although multiple studies have built on the Neoproterozoic large DOC scenario (e.g. Sperling et al., 2011; Cañadas et al., 2022). Purported high oceanic DOC concentrations during the Ediacaran period have been estimated from the Ge/Si ratio of diagenetic chert nodules (Xing et al., 2022) but they reflect the sediments porewater geochemistry and remain difficult to directly relate to the ocean water itself. Besides, some modeling approaches have suggested that DOC abundance in the past Earth's oceans could not have markedly differed from today's values (Fakhraee et al., 2021).

Modern analogous systems such as the Black Sea or Mexican lakes studied here support the possibility of greater DOC accumulation in anoxic waters (Ducklow et al., 2007), but only to levels substantially lower than those required to account for the Neoproterozoic CIEs (minimum concentrations estimated between 25 and 100 mM; Ridgwell and Arndt, 2015). One could argue that the development of larger DOC pools in the three Mexican lakes from the SOB is hindered by relatively large sulfate reservoirs (especially in Alchichica ~10 mM). However, we notice that La Alberca does not show a larger DOC reservoir despite having the lowest oxidant availability (both oxygen- and sulfate-free at depth) and being the only one of the four lakes to present isotopic signatures associated with methanogenesis (Havas et al., 2023). Furthermore, the Mexican lakes are seasonally oxidized, which could consume part of their DOC reservoir. However, the Black Sea is permanently stratified and shows even lower [DOC], suggesting that DOC production might be the primary control on DOC concentration over DOC oxidation. The processes of DOC production and accumulation in the Neoproterozoic ocean could have been less efficient than today (Fakhraee et al. 2021). Nonetheless, an important limit to the analogy between modern analogues and the Precambrian oceans is the difference in time over which DOC could have accumulated in both environments (Ridgwell and Arndt, 2015). One could expect the formation of such a large autochthonous DOC reservoir to increase the ocean inorganic C isotope composition, by mass balance. However, from $\delta^{13}C_{Carb}$ data compilation (e.g. Fike et al., 2006; Saltzman and Thomas, 2012; Li et al., 2017), we see that there are no positive increases of $\delta^{13}C_{Carb}$ at the magnitude of the negative CIEs tens to hundreds of million years before the Neoproterozoic CIEs. Thus, even if the "oxidant paradox" may have found satisfactory explanations, the origin of the massive DOC reservoir required to generate these excursions still remains to be elucidated (Jiang et al., 2010; Lu et al., 2013; Li et al., 2017).

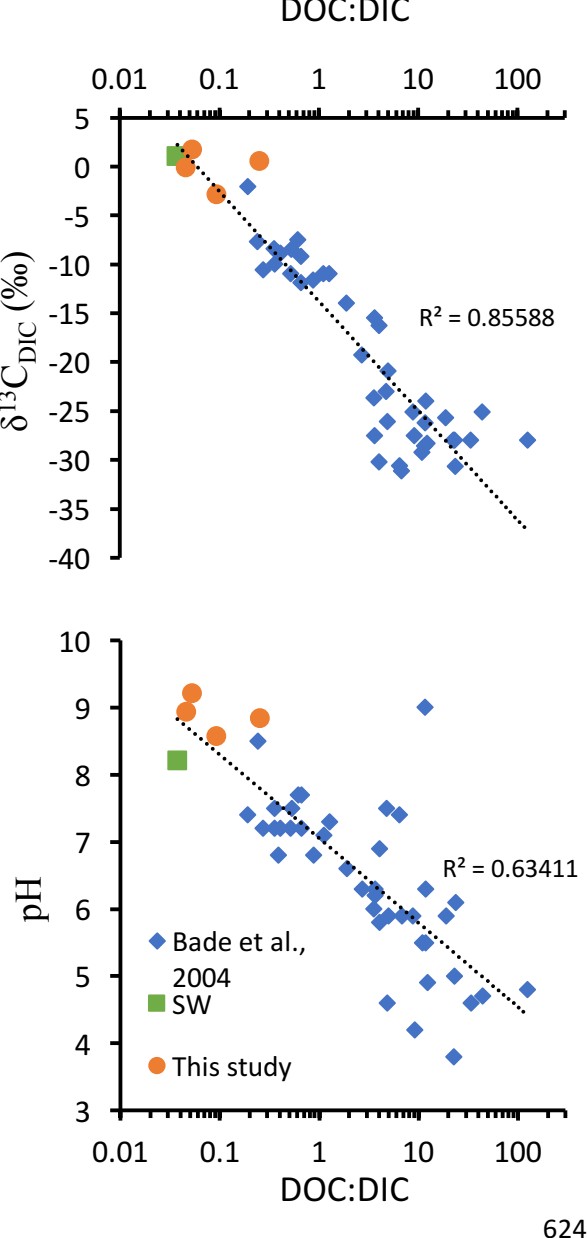

Figure 6. DOC:DIC ratios, pH and $\delta^{13}C_{DIC}$ values from different lakes compiled from Bade et al. (2004) and the four Mexican lakes from Havas et al. (2023), as well as modern surface ocean values (from Kroopnick, 1985; Zeebe and Wolf-Gladrow, 2009 and Hansell, 2013).

Top: $\delta^{13}C_{DIC}$ as a function of DOC:DIC ratio represented with a logarithmic abscissa scale and logarithmic trend line which combines the three datasets.

Bottom: pH as a function of DOC:DIC ratio, with a logarithmic abscissa scale and logarithmic trend line which combines the three datasets.


In the alkaline lakes studied, oxidation of the DOC reservoir would generate a maximum $\delta^{13}C_{DIC}$ deviation of -
2 ‰, in La Alberca de los Espinos, which has the lowest alkalinity. The other lakes $\delta^{13}C_{DIC}$ are less impacted,
notably because they are largely buffered by high DIC content (Havas et al., 2023). Bade et al. (2004) showed that
modern low alkalinity/low pH lakes generally show more negative $\delta^{13}C_{DIC}$ (down to ~ -30 ‰), partly due to a
higher responsiveness of the $\delta^{13}C_{DIC}$ to remineralization of OM and especially DOC. Compiling our data with
those of Bade et al. (2004), we consistently show a clear negative trend of $\delta^{13}C_{DIC}$ with an increasing DOC:DIC
ratio over a broad range of lacustrine DOC and DIC concentrations (Fig. 6). This trend also matches modern ocean
values (Fig. 6). These observations are consistent with the inference that systems where DOC:DIC >> 1 should
drive $\delta^{13}C_{DIC}$ to very negative values (Rothman et al., 2003). However, in modern environments, the biomass is
largely influenced by aerobic heterotrophs and high DOC:DIC waters usually lean toward acidic pHs (Fig. 6; Bade
et al., 2004), at which carbonate precipitation is prevented. Instead, in anoxic waters, remineralization of OM
through sulfate- or iron-reduction generates alkalinity (e.g. Tziperman et al., 2011). Hence, environmental

conditions where DOC:DIC >> 1 might be inconsistent with large carbonate deposits unless they are associated with anaerobic remineralization. This further supports the hypothesis that negative $\delta^{13}C_{Carb}$ excursions of the Ediacaran were triggered by continental sulfate addition to the ocean (Li et al., 2017; Shields et al., 2019; Chen et al., 2022), but following the oxidation of DOC by anaerobic (e.g. sulfate reduction) rather than aerobic (e.g. by free oxygen) pathways. At the same time, additional DOC inputs (e.g. terrigenous) might be necessary to reach the required high DOC conditions allowing the Neoproterozoic CIEs. This echoes previous suggestions of "Neoproterozoic greening", referring to a phase of biological land colonization, although evidence for this phenomenon currently remains equivocal (Lenton and Daines, 2017). While a concomitant supply of sulfate and DOC *via* rivers may cause – at least – a partial oxidation of DOC, it would still result in a $^{13}C$-depleted source of alkalinity to the coastal environments.

The inferences from Fig. 6 also foster the scenario proposed by Tziperman et al. (2011) where the anaerobic respiration of a large DOM production leads to the sequestration of newly produced C in carbonates – with very negative $\delta^{13}C$ – and thereby to the drawdown of atmospheric $pCO_2$ and the initiation of Cryogenian glaciations. We therefore suggest that the climatic feedbacks associated with the negative Neoproterozoic CIEs have been controlled by the total amount and balance between different DOC sources (autochthonous vs. allochthonous), and different oxidation pathways (e.g. via $O_2$ vs. $SO_4^{2-}$).

In summary, Neoproterozoic carbonate carbon isotope excursions likely require DOC and DIC pools to be spatially decoupled (e.g. through terrestrial DOM inputs), which suggests that DOC was not necessarily larger than DIC in the entire ocean. The analogues studied here further support that the Neoproterozoic CIEs recorded in carbonates should have occurred following DOC oxidation through anaerobic rather than aerobic pathways.

## 6. CONCLUSIONS AND SUMMARY

Based on its concentration and isotopic signatures, we characterized the nature and role of the DOC reservoir within the C cycle of four stratified alkaline crater lakes, in comparison with previously described DIC and POC data. Despite similar contexts, the DOC reservoirs of the four lakes show considerable variability, driven by environmental and ecological differences, as summarized below:

- The DOC is the largest OC reservoir in the water column of the studied lakes (> 90%). Its concentration and isotopic composition provide novel information about the C cycle of these stratified water bodies. In each of the four lakes, diverse photosynthetic planktonic communities release greater or smaller amounts of DOC, depending strongly on environmental factors such as nutrient and DIC availability, and transfer most of the inorganic C to DOC rather than POC.
- This process is marked by very heavy and distinct isotopic signatures of DOC compared with POC. They reflect different metabolism/C fixation pathways and/or the activity of a DIC-CM coupled with an overflow mechanism (i.e. DOM exudation), which could be active for both oxygenic and anoxygenic phototrophs, and for which we propose a novel isotopic model of cell carbon cycling, integrating DOC molecules.
- The DOC reservoir in one of the lakes was not characterized by this release process, but rather by partial degradation and accumulation in anoxic waters, associated with more negative isotopic signatures.

–   Our results bring further constraints on the environmental conditions under which autochthonous DOM
can accumulate in anoxic water bodies, providing boundary conditions to the large DOC reservoir
scenarios. This study of modern redox-stratified analogues supports the idea that a large oceanic DOC
reservoir may have generated the record of successive C isotope excursions during the Eocene. Our study
suggests, however, that the Neoproterozoic large DOC hypothesis and its record in carbonates as negative
CIEs would only have been possible if external DOC sources largely contributed, and if DOC oxidation
occurred *via* anaerobic pathways.

682

**Data Availability**
Data are publicly accessible at: https://doi.org/10.26022/IEDA/112943.

**Author Contributions**
RH and CT designed the study in a project directed by PLG, KB and CT. CT, MI, DJ, DM, RT, PLG and KB collected the samples on the field. RH carried out the measurements for C data; DJ the physico-chemical parameter probe measurements and EM provided data for trace and major elements. RH and CT analyzed the data. RH wrote the manuscript with important contributions of all co-authors.

**Competing Interests**
The authors declare that they have no conflict of interest.

**Disclaimer**

**Acknowledgements**
This work was supported by Agence Nationale de la Recherche (France; ANR Microbialites, grant number ANR-18-CE02-0013-02). The authors thank Anne-Lise Santoni, Elodie Cognard, Théophile Cocquerez and the GISMO platform (Biogéosciences, University Bourgogne Franche-Comté, UMR CNRS 6282, France). We thank Céline Liorzou and Bleuenn Guéguen for the analyses at the Pôle Spectrométrie Océan (Laboratoire Géo-Océan, Brest, France) and Laure Cordier for ion chromatography analyses at IPGP (France). We thank Nelly Assayag and Pierre Cadeau for their help on the AP 2003 at IPGP. We warmly thank Carmela Chateau-Smith for improving the syntax and clarity of the manuscript.

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
