# Peer review of "The hidden role of dissolved organic carbon in the biogeochemical cycle of carbon in modern redox-stratified lakes"

_EGUsphere, 2023_

## Author Response (AR1)

In this document, we present our response to the three reviews that were provided on our manuscript "The hidden role of dissolved organic carbon in the biogeochemical cycle of carbon in modern redox-stratified lakes". First, we would like to thank the reviewers for accepting this task and providing helpful comments to improve our manuscript, as well as the associate editor Tina Treude.

The comments of the reviewers are in black and the line numbers they indicate are those from the manuscript initially submitted). We have answered each of the reviewers' comments in blue, and the line numbers are from our modified manuscript (unless specified).

The two main modifications brought to the initial manuscript are the addition of (i) sufficient context and information in that paper such it can be read independently (notably without having read the companion manuscript), and (ii) a clear presentation of our interpretations' limits in the discussion. After corrections following the reviewers' comments, the text was fully reread by a colleague who is a native speaker of British English to correct the problematic grammar and phrasing.

Referee N° 1

The manuscript by Havas et al., entitled "The hidden role of dissolved organic carbon in the biogeochemical cycle of carbon in modern redox-stratified lakes" presents data from four endorheic, alkaline lakes in Mexiko. This manuscript is the result of a split of a much larger previous one into two parts. The other part was also submitted to Biogeosciences and deals with the carbon isotopes of these lakes.

Overall, this manuscript contains interesting and high-quality data that justify publication in Biogeosciences. However, in its present state the manuscript is not ready for publication. There is a general imbalance in the manuscript regarding the lengths of the different parts. The manuscript relies strongly on the other manuscript (Havas et al., submitted) from the split of the larger one, which is currently also under consideration in Biogeosciences. While it is OK to keep things succinct, every manuscript has to stand on its own and should be understandable without any additional literature. This is clearly not the case here.

We thank the reviewer for reading and commenting on our initial manuscript and the new subdivided manuscripts.

We agree that our two manuscripts must stand on their own, as mentioned by the other reviewers. We have therefore added the main data about concentration and isotopic compositions of DIC and POC in the results section, and we have added specific DIC/POC information where required in the discussion, to better understand our arguments about DOC (see discussion 5.2). The isotopic differences between DIC/POC and DOC ($\Delta^{13}C$ or $\varepsilon^{13}C$) are present in the second manuscript in table 2 and figure 4, and thus available for the reader.

We think that the manuscript can now be read independently, as it contains all the information needed to stand on its own.

The introduction is extremely short, and lacks a clear statement of the motivation for this study. Moreover, there are parts scattered throughout the manuscript that should rather go into the introduction, e.g. lines 250-254.

The study of DOC and its stable isotopes is scarce in stratified environments, yet it could bring valuable new information about the C cycle, notably on the reactions related to the production and recycling of DOC. The motivation for our study is to investigate how DOC isotope signatures vary, in relation with specific geo-physico-chemical and ecological factors, all of which are well characterized in the four stratified systems studied (e.g. Zeyen et al., 2021; Iniesto et al., 2022; Havas et al., submitted). To better present the stakes of our work, we have added several sentences describing the state-of-the-art about DOC isotopes (line 57-66). In lines 86-96, we now state more clearly (i) the validity of our dataset to fill in this literature, (ii) how we use it to discuss the mechanisms of DOC production and recycling, and how they are influenced by ecological/environmental constraints, (iii) how these results complement and deepen our understanding of the C cycle in the four lakes compared to more classical DIC/POC analysis, and (iv) how these modern analogues are well suited for a novel exploratory approach to the hypothesis of a past large oceanic DOC reservoir and related C isotope excursions.

As recommended by the reviewer, parts of the discussion have been moved to the introduction, including lines 250-254 and 257-260 (in the initial manuscript).

The Results section is also extremely short and does not present all data. Still, I really liked the description of the results from each of the lakes, as it summed up the major findings in a single paragraph. However, I think this section would improve by providing a clear take-home message for each lake. The current manuscript stops half-way in that respect and still gets lost sometimes by describing too much detail in the text. Figure 4 is not mentioned once in the Results, but only in the discussion, why is that? This figure clearly presents results.

The results section focuses on the main DOC data (i.e. [DOC], $\delta^{13}C_{DOC}$, $\Delta^{13}C_{DOC-POC}$, $\Delta^{13}C_{DOC-DIC}$). As requested in previous comments and by the other reviewers, direct information about DIC and POC data has been added in the results section. To complement this section, we have also added a description of the chlorophyll, oxygen depth profiles, and of the main nutrient concentrations (lines 190-194; 205-208; 219-222; 234-237), all of which are used in the discussion. Figure 4 is called when we mention the $\Delta^{13}C_{DOC-POC}$, $\Delta^{13}C_{DOC-DIC}$ data. All data are now mentioned in both the results and discussion sections.

The discussion is excessive and I have a hard time following the author's claims about the Neoproterozoic and Phanerozoic carbon perturbation events. In my opinion the authors massively over-extrapolate their data, and I would recommend to cut these parts out and simply focus on the modern lakes. There are enough good data to make this a nice round story without overselling.

As mentioned in comments by the other reviewers, there are several limitations to the analogy between ancient oceans and modern lacustrine analogues. We have addressed these remarks by clearly underlining these limits and tempering our three main conclusions (lines 483 and 497-503; 529-530 and 533-535; 563-566). We show that the DOC levels required in the Eocene hypothesis are attained in the seasonally oxidized environments of the four lakes studied, although the scale factor still needs to be assessed. We show that modern analogues favoring the accumulation of DOC (Mexican lakes, Black Sea) are still far from the DOC levels needed for the Neoproterozoic events; thus we suggest that negative C isotope excursions (CIEs) may have required the addition of an external DOC source. Combining our dataset with modern ocean and other lacustrine data supports the hypothesis that anaerobic DOC oxidation pathways would be required to account for the Neoproterozoic CIEs.

We believe that our study of modern analogues provides an alternative and original approach, and a first attempt to directly test the plausibility of hypotheses regarding a large [DOC] in ancient oceans, whereas most studies assume its *prima facie* reality, although no direct demonstration of its existence has so far been given. The study of these four Mexican lakes, with their specific environmental contexts, helps to constrain the parameters allowing DOC to accumulate in modern stratified environments, and to identify the pathways through which it may transfer its isotopic signature to the carbonate record. We thus think that this part is a valuable addition to the manuscript.

Overall, this is a very solid dataset, but the manuscript needs a major revision prior to publication.

We thank the reviewer for this positive assessment of our study and are confident that this version, revised to take into account all the reviewers' remarks (including the addition of relevant data from the companion paper), will provide a much more clearly understandable message, allowing this manuscript to stand on its own.

Referee N° 2

This manuscript by Havas et al. presents and discusses a comprehensive dataset of DOC concentrations and $\delta^{13}$C values from four redox-stratified Mexican lakes. They hypothesize peaks in DOC concentration are due to direct release by photosynthetic organisms, and that DOC $\delta^{13}$C values uniquely record specific metabolic processes in phytoplankton. They then use these results to evaluate the possibility that a large DOC reservoir could explain past C isotope excursions. The manuscript is well written, thorough, and provides novel contributions to use of DOC and $\delta^{13}$C to understand past environmental conditions. However, there are some changes which could improve the manuscript.

Notably, this manuscript appears to be a companion article to an additional study completed at the same time and on the same samples, Havas et al. (Submitted). While it is reasonable for the two manuscripts to be submitted separately, I feel that in some places the authors rely too heavily on the reader having already read Havas et al. (Submitted). I think it would be helpful if the authors were to explain early in the manuscript that these are two separate studies completed on the same samples and what data and major conclusions from the other paper are important for understanding this manuscript.

We thank the reviewer for reviewing our manuscript, and providing this positive appreciation. We agree that this manuscript relies on a previous companion paper on multiple occasions. Following the above general recommendations and specific reviewer's comment bellow (e.g. lines 75-82), we explicitly state in the introduction (line 86-89) that the new DOC data were measured from the same water samples and are thus comparable with previous data from Havas et al. (submitted). We indicate where necessary the data from the companion paper used for comparison.

**Specific comments:**

Line 72: It is mentioned how few studies have measured DOC $\delta^{13}$C values, but does not give any context here or earlier as to why $\delta^{13}$C values are informative and should be measured.

We agree that this aspect was lacking in the introduction. We have added a thorough description of existing literature about DOC $\delta^{13}$C values (line 57-66).

Lines 75-82: This could be a good place to be more specific regarding new data in this manuscript vs. what is published elsewhere. The authors state what data they are reporting as new (concentrations and isotopic compositions of DOC) and list many other publications with background data on those sites. However, I think it would be helpful if they specifically note here that these are the same water samples used in Havas et al. (Submitted) and what data is used as context from Havas et al. (Submitted) (they noted DIC and POC measurements, but based on figure 2, it appears much more than that).

Additionally, it unclear what analyses are referred to by "physico-chemical characteristics" without looking up the referenced manuscripts. Are Vilclara et al. (1993) and Zeyen et al. (2021) examples of the types of analyses referred to by "phyisco-chemical characteristics", or is data from those papers compared to as well as Havas et al. (Submitted)?

We thank the reviewer for this suggestion and we have indicated that DOC and DIC/POC as well as the water physico-chemical characteristics were measured from the same lakes and water samples. To clarify that the new DOC dataset is compared to specific pre-existing data from Havas et al.

(submitted), we have removed the two other citations from that sentence. We also specify which "physico-chemical" parameters we rely on.

Line 185: $\delta^{13}$C DOC was "mostly around"- could the authors be more specific? Ex: $\delta^{13}$C DOC was ± 0.5 ‰ from -26 ‰...

We have modified the text accordingly: "The $\delta^{13}C_{DOC}$ was -25.9 ± 0.4 ‰ throughout the water column except…" (Line 198).

Line 245-246: The figure reference implies there is DIC And POC data in Fig. 3, which there is not.

The reviewer is correct. We have moved the reference to Fig. 3 to the end of the sentence about DOC. The reference to Havas et al (submitted) is only for DIC/POC data.

Line 248: Please provide a statistic and/or reference to where data can be found for the lack of correlation between [DOC] and salinity.

We have added R² and p values for the correlation between average salinity and DIC (R²=0.93, $p$=5.8*10$^{-5}$) and DOC (R²=0.47, $p$=0.2) values (Lines 254-256).

Line 277: In which lakes was the peak in oxic vs. anoxic waters? In section 5.2 it is mentioned it is possible there is an OM increase in anoxic waters due to better preservation, is this the only lake where the peak occurred in anoxic waters?

In Lake La Preciosa, a DOC peak is found only in the oxic waters. In Lake La Alberca, one peak is found in oxic and another in anoxic waters. In lakes Atexcac and Alchichica the DOC peaks are only present in anoxic waters (the increase starts in the oxycline for Atexcac).

We mention the preservation and accumulation of DOC in the anoxic waters of Alchichica in section 5.1.2. Alchichica is not the only lake where DOC increases in anoxic waters, but it is the only one where the increase could not be directly related to autotrophic production (as opposed to Atexcac and La Alberca). Therefore, the origin of this hypolimnetic DOC was interpreted as reflecting partial degradation (notably of POC) and accumulation in anoxic waters, preserving more recalcitrant DOM.

Line 312: Unclear what "different origins" refers to. DOC and POC have different origins? Or the origin of the offset is something other than a switch from CO2 to HCO3?

We agree that the phrasing was unclear. The "different origins" refer to what follows in the text, that is, the different possible explanations for the offset between $\delta^{13}C_{DOC}$ and $\delta^{13}C_{POC}$. We have modified the text to make it clearer (Lines 316-343).

Line 376: This paragraph (and much of the discussion above) seems to assume the reader already knows that the studied lakes have low CO2 vs HCO3 and local competition for CO2. This seems important for the discussion regarding C isotope fractionation during photosynthesis, but this is the first time it's mentioned. Please elaborate on this data either in the site description (which has a brief mention of alkalinity, but no specifics) or earlier in the discussion.

We agree that this information is lacking prior to this part of the discussion. Following the reviewer's suggestion, we have added a brief notice of this important factor in the site description section: "Under these conditions, DIC is composed of $HCO_3^-/CO_3^{2-}$ ions with minor amounts of $CO_{2(aq)}$ (< 0.5 %)." Line 114.

Line 394: A few summary/concluding sentences would be helpful for the reader.

We agree that the preceding discussion is quite dense and a summary would help the reader identify the take-home message. We have added a short summary, lines 364-369.

Section 5.1.1: nowhere in this section do the authors discuss the possibility of degradation of the hypothesized freshly produced OM by heterotrophic bacteria. At line 271, they note that oligotrophic conditions tend to inhibit heterotrophic bacterial activity and suggest this could account for the trend in DOC concentrations (lower concentrations in less oligotrophic lakes due to heterotrophic bacterial degradation), implying heterotrophic degradation should be considered. At the same time, in line 423, they note that negative 13CDOC-POC values would be expected from heterotrophic degradation of DOC, while all lakes discussed in section 5.1.1 had positive offsets. These two ideas seem to contradict each other. Please address the possibility of heterotrophic bacterial degradation in section 5.1.1 and any implications it may have for the conclusions in this section.

We have edited and clarified section 5.1.1. Because $\Delta^{13}C_{DOC-POC}$ shows (strongly) positive values for the first 3 lakes but negative ones for Alchichica, we propose different mechanisms as the origin of their respective DOC reservoirs. In section 5.1.1, we discuss how autotrophic DOC release could generate such positive $\Delta^{13}C_{DOC-POC}$ values. We have also addressed bacterial heterotrophy and photo-degradation, reaffirming why the very positive $\Delta^{13}C_{DOC-POC}$ are not consistent with these processes (lines 310-314).

Line 426: It is unclear what this is referring to in Havas et al. (Submitted), but based on other references to this paper, it appears Havas et al. (Submitted) does not include $\delta^{13}C$ values of more labile vs refractory DOC molecules. Please reference more relevant papers which support heavier isotopic values for labile biomolecules.

We have replaced Havas et al. (submitted) with more relevant references (line 397-399).

Line 427: Please be more specific regarding what in the DIC and POM data was consistent with heterotrophic activity for readers who have not read the other paper.

The $\delta^{13}C_{DIC}$ of Alchichica decreased toward the bottom waters while $\delta^{13}C_{POC}$ increased, as expected from bacterial heterotrophic activity, which transfers light C to the DIC reservoir, while the bacterial biomass (POM) would become $^{13}C$-enriched by consuming $^{13}C$-rich labile organic molecules. The decreasing C:N$_{POM}$ ratio is also consistent with that process.

We have added the required DIC and POM information from Havas et al. (submitted) as requested (line 400-402; together with numerical values added in the result section).

Line 433: Unclear what this means. Was there a shift to cyanobacterial DOC found predominantly in the hypolimnion vs. other parts of the water column? Or did DOC in the hypolimnion shift to predominantly cyanobacterial origin vs. other sources?

We have now edited this sentence to clarify: "A shift of the cyanobacterial DOC from the epilimnion toward the hypolimnion of Lake Alchichica was described at the end of the spring (Alcocer et al., 2014a; 2022)". (Lines 407-408).

Lines 444-450: It's unclear exactly what the point of the paragraph is and how it relates to the section below. Perhaps it is more relevant to the start of section 5.3, explaining the large size of the DOC pool in these lakes vs. today's open ocean?

We thank the reviewer for this remark. We have moved this paragraph to the start of section 5.3.

Line 474: would be helpful to reader if this section ended with one summary sentence regarding the information potential of DOC $\delta^{13}C$ values vs. POC or DIC.

We have added a summary sentence as suggested.

**Figures:**

Fig. 2 caption: It is noted that the epi, meta, and hypolimnion layers are visually represented, but not described how. Please be specific that these are represented as grey shading in background (as noted in the main text)

We specified, as written in the text, that "epi, meta, and hypolimnion layers are represented for each lake by the white, gray, and dark gray areas, based on temperature profiles with the metalimnion corresponding to the thermocline".

Fig. 3: Do these grey boxes correspond with the same depths as in Fig. 2? It is noted in Fig. 2 the grey boxes usually (but not always?) correspond with oxygen rich vs poor conditions. If these are the same, please note this. If not, it is somewhat confusing.

The shaded layers in Fig. 3 (and 4) indicate the same depths as in Fig. 2. They were drawn according to the thermal stratification in each lake (i.e. following the temperature vertical profiles) as indicated in the text. The confusion arises from the fact that the resulting epi-, meta- and hypolimnion layers correspond well to oxygen-rich, intermediate, and oxygen-poor conditions in three of the lakes, but not as closely for La Preciosa, where the oxycline layer is thinner than the thermocline (~5 vs 8 m).

To clarify this, we have added a sentence in the text: "… and correspond to the oxygen-rich, intermediate, and oxygen-poor layers in the four lakes, although the oxycline in La Preciosa is slightly thinner than the thermocline (~5 vs. 8 m)." lines 175-177. Additionally, we also clarified this point in the caption of Fig. 2 (see comment above) and refer to that figure in subsequent figure captions.

Fig. 3 & Fig. 4: it might be easier for the reader to visualize down water column trends if there were a line through the points from surface to deep.

Since these data are discrete analyses and not continuous or with a very tight sampling step, we think that the trend line is not pertinent here and will neither help the visualization of the data nor their interpretation.

Fig. 4 caption: should mention POC and DIC data for this calculation is from Havas et al. (submitted). I think it would be clearer if it read "Vertical profiles of the difference in $\delta^{13}C$ values of DOC and DIC (top)…"

We have added the reference to Havas et al. (submitted) and we changed the phrasing as recommended.

**Technical corrections:**

Line 151: Please be consistent with the number of decimal places and/or significant figures throughout the manuscript (here and in results).

We thank the reviewer for this remark. We have homogenized the significant figures throughout the manuscript.

Line 286: "nature of" not "nature pf" We have corrected the typo.

Lines 588-590: "Depending on the lake" is redundant considering the sentence begins with "Depending on environmental factors…" We have modified the text.

Referee N° 3

The authors introduce a set of geochemical datasets measured on dissolved organic carbon (DOC) obtained from four redox-stratified lakes in Mexico. They compare these datasets with corresponding data obtained from particulate organic carbon (POC) and dissolved inorganic carbon (DIC). Through their analysis, the authors discovered that the concentrations and isotopic compositions of the DOC exhibit significant variability both between and within the lakes. This variability is attributed to differences in the origins of the DOC, which are associated with primary productivity linked to oxygenic and/or anoxygenic processes, redox conditions, and an old long-term DOC reservoir. The authors further extrapolate their findings to the 'DOC' hypotheses proposed for ancient oceans, including the PETM and Shuram events.

The subject is a topic of interest and of significance to the study of those redox-stratified environments, which, I believe, is well-suited for the *BG* readership. I am convinced that the data presented in this contribution will enhance our understanding of the biogeochemical cycle of carbon in such environments, and will provide a valuable reference for the study of DOC in ancient or future oceans. While I believe that this work deserves to be published, it is not yet ready for publication. From a technical standpoint, the writing in this manuscript contains numerous grammatical errors and is not yet at a journal-ready level. I have identified some of these errors, but there are likely more, and the manuscript must be thoroughly checked for technical soundness before it can be resubmitted. Furthermore, some of the discussion appears to be underdeveloped, and I had difficulty following the logic behind certain key conclusions. I have included some general comments below, as well as specific comments throughout the manuscript, which I recommend that the authors address (or at least consider) in order to improve the quality of their work.

We thank the reviewer for providing such detailed and constructive feedback for our manuscript, and for the positive appreciation of our work. We have addressed all of the general and specific comments made by the reviewer and we thank the reviewer for helping us to improve the quality of our report.

After corrections following the reviewer's comments, the text was reread by a colleague who is a native speaker of British English to correct the problematic grammar and phrasing.

General comments

Upon reviewing this contribution, I noticed that a significant amount of the discussion and conclusions drawn seem to rely heavily on data from an unpublished paper (Havas et al., submitted), which appears to also be authored by the same authors. This manuscript does not provide any information about those data, which makes it difficult for readers to verify and evaluate the validity of the findings. While I understand the authors' decision to split their data into separate articles, I believe they should consider adding more details to this manuscript to provide context and transparency for readers. Please see my specific comments below for suggestions on how to improve the presentation of this information.

As requested by both reviewers, we have added more information about the DIC/POC data from Havas et al. (submitted) where suggested, we have also specified in the introduction that the same samples were analyzed in this and the companion paper. We hope that these modifications will provide more context and transparency for readers.

Another point of concern is the authors' extrapolation of their results to the Neoproterozoic carbon perturbation event. They link the observed DOC concentrations in the studied lakes to the 'big DOC' hypothesis for the Neoproterozoic oceans and suggest that increased terrigenous DOC inputs could have been necessary to generate high DOC: DIC conditions and initiate the Neoproterozoic carbon isotope excursions. That is because the proposed DOC concentration in that hypothesis is much higher than the observed values in these lakes. However, I do not believe that it is appropriate to make a simple analogy between today's anoxic lakes and Precambrian anoxic oceans. For one, the duration of today's lakes in redox-stratified conditions is not on the same timescale as the Precambrian oceans. These lakes are seasonally oxidized, which would consume the DOC that accumulated during earlier times, whereas the Precambrian oceans could have been permanently stratified over much longer timescales (millions of years). Additionally, the authors suggest that terrigenous sulfate input is the main oxidant of the DOC, which raises concerns. During a long river journey, there would be sufficient time for terrigenous DOC oxidation by sulfate and for equilibrium between newly-formed $^{13}$C-depleted DIC and pre-existing 'normal' $CO_2$. It is unclear whether the $^{13}$C-depleted signatures could be transported into the oceans and preserved in the marine facies.

We thank the reviewer for these insightful and constructive remarks. While we agree there are limitations to the proposed analogy, we think that several aspects of that comparison can bring new insights about the geological intervals mentioned, and thus should be discussed. Discussing these points, as recommended by the reviewer and as described below, will better present the limits of our proposition while expanding its horizons.

We agree that a major drawback in this analogy lies in the fact that the time during which DOC can accumulate in these modern stratified analogues and the Precambrian ocean differs significantly. Hence, we have added a clear cautionary statement in that sense (lines 533-535), and we have justified how this difference may or may not explain the discrepancy between the lakes' [DOC] and purported Precambrian [DOC]. We note that our discussion is a first attempt to test the plausibility of such large [DOC] in the Precambrian ocean, whereas most studies assume its *prima facie* reality, although no direct demonstration of its existence has so far been provided.

We agree that the monomictic nature of the lakes studied is another important difference with the Precambrian oceans. As we indicate in the text, the Black Sea, a permanently stratified basin, harbors even lower DOC concentrations, suggesting that the mechanisms producing the DOC might be the primary controls on DOC concentrations, despite differences in conservation potential (more or less DOC oxidation). Nevertheless, this is indeed a limitation to the suggested analogy with the Precambrian oceans, which were permanently stratified over much longer timescales. Hence we now address this point more clearly in the text (lines 529-533).

We do not precisely know the kinetics of DOC oxidation by sulfate reduction in rivers during the Neoproterozoic, but it would have been strongly dependent on the amount and lability/refractory nature of DOC that is being oxidized, as well as the distance between the continental source and the ocean. These parameters cannot be constrained but they would not necessarily lead to a full oxidation of the putative terrestrial OC pulse. Based on the previous discussion, we show that there is still no satisfactory explanation for the existence of a large Precambrian DOC reservoir (Line 539-541). Hence, terrestrial DOC appears as a potential source for this DOC, and is consistent with other pulses of continental supply (e.g. sulfates).

In the scenario we propose, it is indeed possible that at least part of the DOC is oxidized during the concomitant journey of DOC and sulfate in rivers. We now address this possibility in lines 564-566. Importantly, the requirement for generating the negative Neoproterozoic CIEs is to have a major influx of $^{13}$C-depleted DIC. In the "big DOC" hypothesis, this is achieved *via* ocean DOC:DIC >> 1, and partial

oxidation of the DOC reservoir. In our proposition, allochthonous (e.g. terrigenous) supply is necessary to reach the required amounts of OC. Nonetheless, a partial (or even full) anaerobic oxidation of that OC before it reaches the ocean would allow the CIE to be recorded, as it still provides a source of $^{13}$C-depleted alkalinity, but would not change the conclusion of a decoupling between DI$^{13}$C-depleted and DI$^{13}$C-enriched pools.

Specific comments

Line 105: Please add a 'during'. Done.

Line 118: Double 'is', please delete one. Done.

Line 140: Before introducing the measurements of DOC concentration, you should provide information on how you get the DOC first.

The DOC concentrations and isotope compositions were measured on the bulk DOC. Therefore, there was no specific step of DOC extraction; it was directly analyzed from the acidified water. We now provide this information, and state what "bulk DOC" refers to in lines 152-154: "…to degas all the DIC and leave DOC as the only C species in solution. The bulk DOC was analyzed directly from the acidified waters (i.e. all organic C molecules smaller than 0.22 μm)".

Lines 147-148: Are you sure the gas was separated in a reduction column? I thought it should be in a GC column.

In the IsoToc device, in addition to water and halogen condensers, the main separation occurs in a reduction column filled with copper. It reduces the oxidized products other than $CO_2$ (mostly NOx compounds to $N_2$) and removes the excess $O_2$ from the combustion. We have added the reference for the company manufacturing the IsoTOC (Line 158).

Line 161: Please define the 'DOC', namely tell the readers what is the 'DOC' you referred here, before this sentence.

The meaning of DOC is now clearly defined at the beginning of the introduction and in the method section (lines 48-50/152-154).

Also linked to figure 3; I would suggest to keep the order of lake consistent between descriptions in "Results" and the figure 3. From left to right, Alchichica, Atexcac…; or reorder them in results.

We have made the results section and figures consistent with the order of the lakes.

Further, I don't think the brown sediments are necessary, and the authors can also consider plotting DOC and δ$^{13}$C$_{DOC}$ in one column, as well as for Δ$^{13}$C$_{DOC-DIC}$ and Δ$^{13}$C$_{DOC-POC}$. I would also suggest to merge figures 3 and 4 into one figure, with its panel arrangement just like as the figure 2. I believe doing the above will help the readers compare the data from different lakes clearer and more easily.

We prefer to keep figures 3 and 4 separate, because combining [DOC] and δ$^{13}$C$_{DOC}$ data, which are different types of data, in a single graphic column would induce confusion and overload the graphic column of each lake. In figure 4, the graphs are plotted on very different scales, and combining them would obscure the visibility of the variations for each lake and between the lakes. We also feel that

representing the sediment level at the base of each lake (in brown) provides a better visualization of the different water column depths in the four lakes.

Lines 165-167: How did you get the concentrations and isotopes of total carbon; the associated information was not found in the 'Method', please add some details.

Good to find a little information about how the $\delta^{13}C_{Total}$ was obtained in the table 1 caption; I suggest to present it in the main-text as well.

We agree with the reviewer that this part was missing in the previous version. We have thus added it in the method (lines 167-169).

Line 180: The authors did a comprehensive comparison of the data between DOC and POC or DIC in detail, but not even provide any information about the latter, like what are the values of concentrations or isotope compositions of POC or DIC and how to get them. Given much of discussion is based on the data from POC and DIC, I would suggest the authors to add some essential information about the POC and DIC.

The POC and DIC isotope data from Havas et al. (submitted) are only used in the current manuscript to provide a useful comparison with the $\delta^{13}C_{DOC}$. They are only discussed relatively to each other, *via* $\Delta^{13}C_{DOC-DIC}$ and $\Delta^{13}C_{DOC-POC}$ parameters.

We agree that basic information about $\delta^{13}C_{DIC}$ and $\delta^{13}C_{POC}$ data would provide more context for the reader. We have therefore added a short description of these parameters in each result section (lines 188-190; 204-205; 218-219; 232-234) and additional numerical information from Havas et al. (submitted) where needed in the discussion (e.g. section 5.2.).

Lines 189-190: Syntax error for this sentence, please revise it.

We have rephrased that sentence as follows: "The $\Delta^{13}C_{DOC-POC}$ values decreased from ~1.3 ‰ in the upper waters to ~ -0.4 ‰ in the bottom waters but showed a peak to +7.1 ‰ at a depth of 12.5 m" (Lines 202-204).

Lines 245-246: No DIC or POC data profiles was presented in the figure 3; maybe the authors mean the figure 3 in the referred paper?? Please indicate it clearer.

The reviewer is correct and we have moved the reference to Fig. 3 to the end of the sentence about DOC. The passage has been rewritten (lines 251-253).

Lines 247-248: Here maybe the authors were trying to say 'DOC' rather than 'DIC'; also, more details about correlation between the average [DOC] and their salinity should be provided, such as adding the associated data or a figure to support the argument.

We have added statistical parameters ($R^2$, *p* value) about these correlations to clarify and support the point made here: there is no significant correlation between DOC content and lake salinity, but there is one between DIC content and salinity (lines 253-255).

Lines 257-259: This sentence should arise earlier, at least before the data descriptions in the 'results'. Also refer to the comment for line 161.

We have moved this sentence about DOC to the beginning of the introduction.

Lines 267-272: I cannot follow the logic here. As I understand it, nutrient-limited condition would suppress not heterotrophic bacterial activity but also the oxygenic photosynthesis, how to result in the fixed C in excess? Also, could you add more details about how much the excess C is, i.e., the actual C/N/P ratio.

On the one hand, C may be fixed in excess relatively to the amount of nutrients available for the biosynthesis of more complex molecules; it thus ends up being released out the cells (e.g. Hessen and Anderson, 2008; Morana et al., 2014). Therefore, more oligotrophic conditions favor higher DOC production. On the other hand, bacterial heterotrophic activity (which partly consumes DOC) is limited by the lack of nutrients (e.g. Dittmar, 2015). Therefore, more oligotrophic conditions also favor a higher quantity of DOC via higher preservation. In consequence, we state that these processes are consistent with the trend observed in our study, where higher DOC concentrations are found in lakes with more oligotrophic conditions. We have rewritten this section to make it clearer (Lines 267-275). Unfortunately, we do not have C/N/P data for the dissolved organic matter.

Line 286: typo, 'of' Ok.

Lines 300-308: Sorry, I cannot follow the reasoning here. Actually, the release rate of DOC is directly related to PER rather than the fraction of DOC over total OM, right? The data, the fraction of DOC over total OM, just reflects a final result of multiple factors, including the release of DOC, an older long-term DOC reservoir (it should exist as the authors argued in the following discussion), or others. So, if the authors try to reach the conclusion of "the extremely high phytoplankton-release rates", I think they should start with an independent evidence for the percentage of extracellular release. Alternatively, it is also practicable to just exclude the possibility that other factors play a role. Otherwise, we could also say the high DOC over total OC ratios are just results of long-term DOC reservoirs in these lakes. Maybe I misunderstood it, but I would suggest the authors make this argument be clearer.

The reviewer is correct that the fraction of DOC over total OC (as we calculated it) integrates multiple factors, including DOC release rates and DOC accumulation over time. We have rephrased the paragraph to indicate that, since PER values as high as 99% have been reported in other systems, a majority of the DOC measured at depths of oxygenic photosynthesis could in theory result from a recent phytoplankton exudation. We also state clearly that PER was not measured in our study (Line 306-308).

Line 315: $^{13}$C-enriched DOC Modified.

Line 319: What does the CCB mean, please define it.

Typo corrected: CBB.

Line 322-325: I don't understand the logic here, how this sentence is linked to the above discussion? Can you elucidate it clearer? Also, I would suggest to include the data of $\varepsilon DOC\text{-}CO_2$ into the figure 4.

We imply that the second option considered to explain the very positive $\Delta^{13}C_{DOC\text{-}POC}$ (namely, the production of less negative $\delta^{13}C_{DOC}$ via the utilization of a C fixation pathway different from the CBB) could work for the case of La Alberca but not for Atexcac. We have rephrased the sentence to make it clearer (lines 325-330).

The $\varepsilon DOC\text{-}CO_2$ and $\Delta^{13}C_{DOC\text{-}DIC}$ parameters differ from each other only by a consistent 10±0.3‰. Thus, representing both in figure 4 would not visually bring new information, but it might possibly obscure

the variability depicted by $\varepsilon DOC\text{-}CO_2$ or $\Delta^{13}C_{DOC\text{-}DIC}$ within each lake, as it would broaden the scale on the abscissa axis. Since we mainly refer to $\varepsilon DOC\text{-}CO_2$ in the text, we now represent only that parameter in Fig. 4, but we report both parameters in Table 2.

Lines 377-378: 'an intracellular', not 'intracellular a'. Indeed, now corrected.

Figure 5 caption: Is the 'c' indeed different from the 'b'?? It looks like to me that they are both a closed-system characterized by low $CO_2$ concentrations. In 'b', DOC and POC are produced concurrently, but in 'c', 'POC' first and then 'DOC'? Further, how to evolve into a closed-system, can you add some explanations? I though such a system may be difficult to form.

The difference between 'b' and 'c' is that, in the second system, in addition to low $CO_2$ concentrations, the lack of nutrients while high photosynthetic rates are maintained would lead to the fixation of large amounts of OC, which cannot be further anabolized and incorporated into the cell biomass (i.e. POC), and which are thus released as DOC. Hence, the fixed OC entering the cell biomass until nutrients are exhausted is formed first, while subsequently fixed OC is released as DOC.

In 'c' the DOC/POC ratio is very high so that, while both evolved from DIC under a closed system, the POC still imprints the fractionation of C fixation, while the DOC produced in a second step evolves toward the initial isotopic composition of the DIC.

As the activation of a DIC-CM is an energy-costly process, it has been thought to reduce the efflux of DIC (i.e. generating a closed system) from the cell back to the external medium (*cf.* lines 349), which would explain why it can lead to very small isotopic fractionations (e.g. Beardall et al., 1982; Iniguez et al., 2020; 'b' in Figure 5).

While these mechanisms deserve to be further explored in dedicated studies, they are simply proposed here as conceptual models to explain the data. We now emphasize this in the text in lines 351-352 and also in a short summary paragraph (lines 368).

Lines 398: It is weird to only dissection one lake, which is not even the one who has the most DOC accumulation.

The outline aims to illustrate the source/production of DOC (5.1.1) and then its fate (partial degradation and accumulation; 5.1.2). The first three lakes are used to illustrate the production of DOC, while the fate (partial degradation and accumulation) is illustrated by the case of Alchichica.

As discussed in 5.1.1, the prominent DOC peaks can only be directly related to autochthonous production in the first three lakes, but we also mention why these peaks cannot be explained by a degradation process (see answer to previous comment). Thus, the autotrophic production of DOC and its isotopic signatures are discussed through the lens of these three lakes. At the beginning of 5.1.2, we expressly discuss why the case of Alchichica may or may not be different. Since it appears that DOC isotopic signatures are more consistent with the bacterial degradation hypothesis (i.e. refractory DOC originating from DOC and POC heterotrophic partial degradation), we use this lake as an example of DOC fate.

We have modified the title of section 5.1.2. to indicate that Alchichica is used to exemplify the possible fate of DOC in the Mexican lakes.

Line 416: It should be 'C isotope'. Corrected.

Line 421: Should be 'consistent'. Corrected.

Line 438: Delete the 'it'. Deleted.

Line 453: Please provide the actual values of POC or $\delta^{13}C$ here. It means the figure 3 in the referred paper?

We have now deleted the reference to figure 3 to avoid confusion. Based on a previous comment we have also added the relevant data about $\delta^{13}C$ from Havas et al. (submitted). While we provide the exact changes in DIC and DOC concentrations, there is no change in [POC] and $\delta^{13}C_{POC}$, as indicated in the text, and therefore no values to provide.

Line 455: 'DOC isotope compositions'. Corrected.

Lines 451-465: Too much discussion derives from "Havas et al., submitted", which makes it impossible to verify and further evaluate.

We have added the relevant values for the DIC/POC data to show how they contribute to our analysis of the DOC data. We have also rephrased the paragraph to make it clearer.

Lines 466-467: Syntax error, please revise it. We have rewritten the sentence.

Line 501: It is weird to include two 'important'. We have rewritten the sentence (Lines 494).

Line 544: I suggest to transfer the unit 'PgC' to mM to help the readers compare it with the above.

We would like to thank the reviewer for this comment because after rereading the paper by Burdige and Komada (2015), we realize that we made a mistake about the unit of that number and the fact that it only considers coastal and continental margin benthic fluxes (water depths <2000m). This latter actually amounts to 180 TgC.yr$^{-1}$. In addition, Burdige and Komada (2015) report a value of 100 TgC.yr$^{-1}$ for water depths > 2000m. Thus, the total DOC benthic flux would be 280 TgC.yr$^{-1}$ or 0.3 PgC.yr$^{-1}$.

While Burdige and Komada (2015) describe the significance of this benthic flux relative to other DOC fluxes to the ocean (e.g. from rivers) and its importance for the specific oceanic recalcitrant DOC reservoir, the benthic DOC flux seems small compared to the entire oceanic DOC reservoir of 660 PgC. This is in part because, in the modern $O_2$-rich ocean, most of the OC from primary production is remineralized into DIC (e.g. Jahnke, 1996: doi.org/10.1029/95GB03525; Burdige and Komada, 2015). Therefore, the argument about the benthic flux is not as strong here.

In contrast, anoxic bottom waters would likely allow a greater DOC flux from the sediments back to the water column (Dadi et al., 2017; Peter et al., 2016: 10.1002/2016JG003425). However, this was not quantified at a global scale. Nonetheless, Fakhree et al. (2021) discuss how the advent of eukaryotes may not have represented such a radical change, as ballasting with metal oxide particles and OC flocculation as particles already in the Neoproterozoic would go against the view of isolated single-cell prokaryotes that would float in the ocean for longer time periods.

In order to simplify and shorten this section, we have decided to remove this argument and delete this part of the discussion.